# ERROR SLICE DISCOVERY VIA MANIFOLD COMPACTNESS

## ABSTRACT

Despite the great performance of deep learning models in many areas, they still make mistakes and underperform on certain subsets of data, i.e. *error slices*. Given a trained model, it is important to identify its semantically coherent error slices that are easy to interpret, which is referred to as the *error slice discovery* problem. However, there is no proper metric of slice *coherence* without relying on extra information like predefined slice labels. The current evaluation of slice coherence requires access to predefined slices formulated by metadata like attributes or subclasses. Its validity heavily relies on the quality and abundance of metadata, where some possible patterns could be ignored. Besides, current algorithms cannot directly incorporate the constraint of coherence into their optimization objective due to the absence of an explicit coherence metric, which could potentially hinder their effectiveness. In this paper, we propose *manifold compactness*, a coherence metric without reliance on extra information by incorporating the data geometry property into its design, and experiments on typical datasets empirically validate the rationality of the metric. Then we develop Manifold Compactness based error Slice Discovery (MCSD), a novel algorithm that directly treats risk and coherence as the optimization objective, and is flexible to be applied to models of various tasks. Extensive experiments on the current benchmark and case studies on other typical datasets demonstrate the effectiveness of our algorithm.

## 1 INTRODUCTION

In recent years, with the enhancement of computational power, deep learning models have achieved significant progress in numerous tasks (He et al., 2016; Devlin et al., 2018; He et al., 2017). Despite their impressive overall performance, they are far from perfect, and still suffer from performance degradation on some subpopulations (Sagawa et al., 2019; Yang et al., 2023). This substantially hinders their application in risk-sensitive scenarios like medical imaging (Suzuki, 2017), autonomous driving (Huval et al., 2015), etc., where model mistakes may result in catastrophic consequences. Therefore, to avoid the misuse of models, it is a fundamental problem to identify subsets (or slices) where a given model tends to underperform. Moreover, we would like to find coherent interpretable semantic patterns in the underperforming slices. For example, a facial recognition model may underperform in certain demographic groups like elderly females. An autonomous driving system may fail in the face of steep road conditions. Identifying such coherent patterns could help us understand model failures, and we could employ straightforward solutions for improvement like collecting new data (Liu et al., 2023) or upweighting samples in error slices (Liu et al., 2021).

Previously, there are works of *error slice discovery* (d'Eon et al., 2022; Eyuboglu et al., 2022; Wang et al., 2023b; Plumb et al., 2023) towards this goal. Despite the emphasis on coherence in error slice discovery, there is no proper metric to assess the coherence of a given slice without additional information like predefined slice labels. On one hand, this impairs the efficacy of the evaluation paradigm of error slice discovery. In the current benchmark (Eyuboglu et al., 2022), with the help of metadata like attributes or subclasses, it predefines slices that are already semantically coherent, and they depict the coherence of a slice discovered by a specific algorithm via the matching degrees between it and the predefined underperforming slices, so as to evaluate the effectiveness of the algorithm. Such practice heavily relies on not only the availability but also the quality of metadata, whose annotations are usually expensive, and may overlook model failure patterns not captured by existing metadata. On the other hand, due to the absence of an explicit coherence metric, current

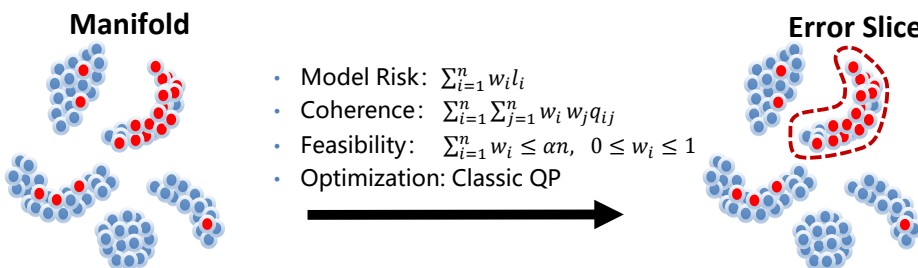

Figure 1: Illustration of Manifold Compactness based error Slice Discovery (MCSD). The blue points are correctly classified by the given trained model, while the red ones are wrongly classified. We can see that the model achieves a good overall accuracy, but exhibit a high error in a certain slice.

algorithms can only indirectly incorporate the constraint of coherence into their design, e.g. via clustering (Eyuboglu et al., 2022; Wang et al., 2023b; Plumb et al., 2023), without treating it as a direct optimization objective. This could potentially impede the development of more effective error slice discovery algorithms.

In this paper, inspired by the data geometry property that high dimensional data tends to lie on a low-dimensional manifold (Belkin & Niyogi, 2003; Roweis & Saul, 2000; Tenenbaum et al., 2000), we incorporate this property to propose *manifold compactness* as the metric of coherence given a slice, which does not require additional information. We illustrate the validity of the metric by showing that it captures semantic patterns better than depicting coherence via metrics directly calculated in Euclidean space, and is empirically consistent with current evaluation metrics that require predefined slice labels. Then we propose a novel and flexible algorithm named Manifold Compactness based error Slice Discovery (MCSD) that jointly optimizes the average risk and manifold compactness to identify the error slice. Thus both the risk and coherence, i.e. the desired properties of error slices are explicitly treated as the optimization objective. We illustrate our algorithm in Figure 1. Besides, our algorithm can be directly applied to trained models of different tasks while most error slice discovery methods are restricted to classification only. We conduct experiments on dcbench (Eyuboglu et al., 2022) to demonstrate our algorithm's superiority compared with existing ones. We also provide several case studies on different types of datasets and tasks to showcase the effectiveness and flexibility of our algorithm. Our contributions are summarized below:

- We define manifold compactness as the metric of slice coherence without additional information. We empirically show that it captures semantic patterns well, proving its rationality.

- We propose MCSD, a flexible algorithm that directly incorporates the desired properties of error slices, i.e. risk and coherence, into the optimization objective. It can also be applied to trained models of various tasks.

- We conduct experiments on the current error slice discovery benchmark to show that our algorithm outperforms existing ones, and we perform diverse case studies to demonstrate the usefulness and flexibility of our algorithm.

## 2 PROBLEM

Unless stated otherwise, for random variables, uppercase letters are used, in contrast to a concrete dataset where lowercase letters are used. Consider a general setting of supervised learning. The input variable is denoted as $X \in \mathcal{X}$ and the outcome is denoted as $Y \in \mathcal{Y}$, whose joint distribution is $P(X, Y)$. There exist multiple slices, where $j$-th slice can be represented as a slice label variable $S^{(j)} \in \{0, 1\}$. For the classic supervised learning, the goal is to learn a model $f_\theta : \mathcal{X} \mapsto \mathcal{Y}$ with parameter $\theta$. Denote $\ell : \mathcal{Y} \times \mathcal{Y} \mapsto [0, +\infty]$ as the loss function. Current machine learning algorithms are capable of learning models with a satisfying overall performance, which can be demonstrated via a low risk $\mathbb{E}_P[\ell(f_\theta(X), Y)]$ over the whole population. However, performance degradation could still occur in a certain subpopulation or slice. Here we introduce the error slice discovery problem:

**Problem 1 (Error Slice Discovery)** *Given a fixed prediction model $f_{\theta_0} : \mathcal{X} \mapsto \mathcal{Y}$ and a validation dataset $\mathcal{D}_{va} = \{(x_i^{va}, y_i^{va})\}_{i=1}^{n_{va}}$, we aim to develop an algorithm $\mathcal{A}$ that takes $\mathcal{D}_{va}$ and $f_{\theta_0}$ as input, and learns slicing functions $g_\varphi^{(j)} : \mathcal{X} \times \mathcal{Y} \mapsto \{0, 1\}, 1 \le j \le K$. Denote the output of $j$-th slicing function as $\hat{S}_j$. We require that the risk in the slice is higher than the population-level risk by a certain threshold: $\mathbb{E}_{X,Y \sim P(X,Y|\hat{S}_j=1)}[\ell(f_\theta(X), Y)] > \mathbb{E}_{X,Y \sim P(X,Y)}[\ell(f_\theta(X), Y)] + \epsilon$, and the discovery slice is as coherent as possible for convenience of interpretation.*

The reason why we require an extra validation dataset to implement error slice discovery is that for deep learning models, training data is usually fitted well enough or even nearly perfect. Thus model mistakes on training data carry much less information on models' generalization capability. This is common practice in previous works (d'Eon et al., 2022; Eyuboglu et al., 2022; Wang et al., 2023b; Plumb et al., 2023). Without ambiguity, we omit the superscript or subscript of "va" for $n, x_i, y_i$ for convenience in the next two sections.

## 3 METRIC

Due to the absence of a proper metric for coherence that is independent of additional information, the current benchmark (Eyuboglu et al., 2022) provides numerous datasets, trained models, and their predefined underperforming slice labels. They employ precision@$k$, i.e. the proportion of the top $k$ elements in the discovered slice belonging to the predefined ground-truth error slice as the metric of slice coherence to evaluate error slice discovery algorithms. Although such practice is reasonable to some extent, its effectiveness of evaluation strongly relies on the quality of metadata that composes the underperforming slice labels, which might be not even available under many circumstances.

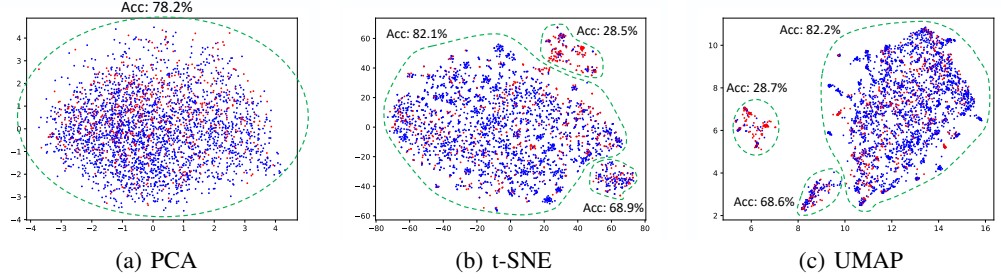

|     |     |     |
| :-: | :-: | :-: |
| (a) PCA | (b) t-SNE | (c) UMAP |

Figure 2: Category "Blond Hair" of CelebA. Visualization of t-SNE and UMAP (manifold-based dimension reduction techniques) shows much clearer clustering structures than that of PCA (mainly preserving Euclidean distances between data points), indicating that it could be better to measure coherence in the metric space of a manifold than using metrics directly calculated in Euclidean space.

To eliminate the requirement of predefined slices, we try to propose a new metric of coherence. It is commonly acknowledged that high-dimensional data usually lies on a low-dimensional manifold (Belkin & Niyogi, 2003; Roweis & Saul, 2000; Tenenbaum et al., 2000). In this case, while direct usage of Euclidean distance cannot properly capture the dissimilarity between data points, the geodesic distance in the metric space of the manifold can. For preliminary justification, here we provide visualization analyses based on different types of dimension-reduction techniques. Among these techniques, PCA mainly preserves pairwise Euclidean distances between data points while t-SNE and UMAP are both manifold learning techniques. In Figure 2, blue dots are correctly classified by the trained model and red dots are wrongly classified. We can see that the visualization of t-SNE and UMAP shows much clearer clustering structures than that of PCA, either having a larger number of clusters or exhibiting larger margins between clusters. This indicates that it could be better to measure coherence in the metric space of a manifold than in the original Euclidean space. Due to the space limit, we only present results of the widely adopted facial dataset CelebA (Liu et al., 2015) here, leaving results of other datasets in Appendix A.1.1, where the same conclusion is true.

Therefore, we attempt to define a metric of coherence inside the discovered slice via the compactness in the data manifold. In practice, the manifold can be treated as a graph $G$ (Melas-Kyriazi, 2020), and we can apply graph learning methods like k-nearest neighbor (kNN) to approximate it (Dann

et al., 2022). Given an identified slice $\hat{\mathcal{S}} = \{(x_i, y_i)|\hat{s}_i = 1\}$, where $\hat{s}_i$ is the output of the slicing function on $i$th sample, we define manifold compactness of $\hat{\mathcal{S}}$ as follows:

**Definition 1 (Manifold Compactness)** *Consider a given approximation of the data manifold, i.e. a weighted graph $G = (V, E, Q)$. The node set $V = \{v_i\}_{i=1}^n$ corresponds to the dataset $\{(x_i, y_i)\}_{i=1}^n$. The edge set $E = \{e_{ij}\}_{1 \le i,j \le n}$, where $e_{ij}$ represents whether node $v_i$ and $v_j$ are connected in the graph $G$. The weights $Q = \{q_{ij}\}_{1 \le i,j \le n}$, where $q_{ij}$ represents the weight of edge $e_{ij}$. Given a slice $\hat{\mathcal{S}}$, the manifold compactness of it can be defined as:*

$$\text{MC}(\hat{\mathcal{S}}) = \frac{1}{|\hat{\mathcal{S}}|} \sum_{(x_i, y_i), (x_j, y_j) \in \hat{\mathcal{S}}} q_{ij} \tag{1}$$

This metric is the average weighted degree of nodes of the induced subgraph, whose vertex set corresponds to the slice. The higher it is, the denser or more compact the subgraph is, implying a more coherent slice. Note that when applying this to evaluate multiple slice discovery algorithms, for convenience of comparison, we control the size of $\hat{\mathcal{S}}$ for those algorithms to be the same by taking the top $\alpha n$ data points sorted by the slicing function's prediction probability. Here $n$ is the size of the dataset and $\alpha \in (0, 1]$ is a fixed proportion. The operation of selecting data points with highest prediction probabilities is akin to calculating precision@$k$ in dcbench (Eyuboglu et al., 2022).

Next, we try to demonstrate the validity and advantages of our proposed coherence metric. A common and representative metric of coherence directly calculated in Euclidean space is variance. Thus we measure variance and manifold compactness respectively on different semantically predefined slices of CelebA (Liu et al., 2015). We use the binary label $y$ to indicate whether the person has blond hair or not, and $a$ to indicate whether the person is male or not. The values of $y$ and $a$ can formulate slices of different granularity. In Figure 3, the most coarse-grained slice is the whole dataset (the darkest circle in the center), the most fine-grained slice is

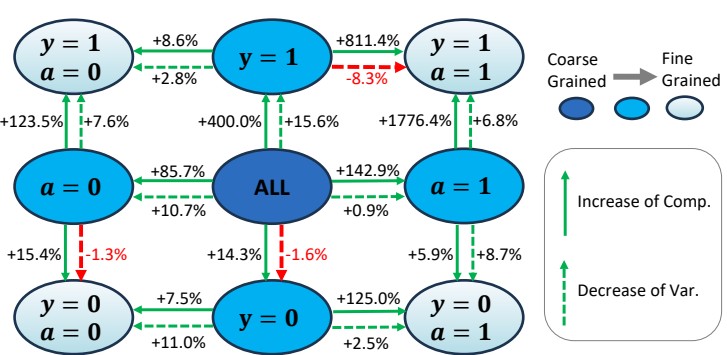

Figure 3: Percentage of increase of manifold compactness ("Comp.") and decrease of variance ("Var.") from coarse-grained slices to fine-grained ones in CelebA. For manifold compactness, there is always a positive increase from semantically coarse-grained slices to fine-grained slices. However, in some cases, variance fails to decrease from more coarse-grained slices to fine-grained slices as expected, which are marked in red arrows. This could imply that manifold compactness is better at capturing semantic coherence than variance does.

the combination of $y$ and $a$ (the lightest circles in the four corners), and slices of the middle granularity are formulated by either of $y$ and $a$. Figure 3 shows the percentage of the increase of manifold compactness and the decrease of variance with directed arrows from semantically coarse-grained slices to fine-grained ones. It is intuitive that these digits are supposed to be positive if these two metrics could properly measure semantic coherence. However, for variance, in some cases the value of the more coarse-grained slice is even smaller than the more fine-grained, marked in red arrows. For manifold compactness, there is always a positive increase from semantically coarse-grained slices to fine-grained slices. In this way, we demonstrate that manifold compactness is better at capturing semantic coherence than variance does. Still due to the space limit, we only provide results of CelebA here, and leave detailed values and results of other datasets in Appendix A.1.2, where we reach the same conclusion. Besides, in Table 1 of Section 5, we have also empirically shown that the rank order of the four methods according to precision metrics is generally the same as that of manifold compactness. Since the precision metrics are based on predefined slice labels with semantic meanings, it implies that our proposed coherence metric could capture semantic patterns well and is appropriate for evaluation of slice discovery algorithms even when predefined slice labels are absent.

---

**Algorithm 1** Manifold Compactness based Error Slice Discovery (MCSD)

**Input:**
Validation dataset: $\mathcal{D} = \{(x_i, y_i)\}_{i=1}^n$.
The trained model to be evaluated: $f_{\theta_0} : \mathcal{X} \mapsto \mathcal{Y}$.
Size of the slice as a proportion of the dataset: $\alpha$.
Coherence coefficient $\lambda$.
A pretrained feature extractor: $h_{fe} : \mathcal{X} \mapsto \mathcal{Z}$.
**Output:** The identified error slice $\hat{\mathcal{S}}$.
**for** $i = 1$ to $n$ **do**
    Calculate the embedding: $z_i = h_{fe}(x_i)$.
    Calculate the model prediction loss: $l_i = \ell(f_{\theta_0}(x_i), y_i)$.
**end for**
Establish the kNN graph $G = (V, E, Q)$ based on the embeddings $\{z_i\}_{i=1}^n$.
Formulate the quadratic programming problem with variables $\{w_i\}_{i=1}^n$ as Equation (2).
Employ Gurobi to solve the problem in Equation (2).
**for** $i = 1$ to $n$ **do**
    $\hat{s}_i = 1$ **if** $w_i > \alpha$-Quantile of $\{w_i\}_{i=1}^n$ **else** 0.
**end for**
**return:** $\hat{\mathcal{S}} = \{(x_i, y_i)|\hat{s}_i = 1\}$

---

## 4 ALGORITHM

We introduce Manifold Compactness based error Slice Discovery (MCSD), a novel error slice discovery algorithm that incorporates the data geometry property by taking manifold compactness into account. In this way, the metrics of both risk and coherence can be treated as the explicit objective of optimization, thus better enabling the identified error slice to exhibit consistent and easy-understanding semantic meanings. The detailed algorithm is described in Algorithm 1. It is worth noting that although we mainly focus on the identified worst-performing slice for convenience of analyses and comparison, our algorithm could discover more error slices by removing the first discovered slice from the validation dataset and applying our algorithm repeatedly to the rest of the dataset for more error slices. Related experiments and analyses are included in Appendix A.2.

First, we approximate the data manifold via a graph. To facilitate the graph learning approach, we obtain the embeddings of the dataset via a pretrained feature extractor (Radford et al., 2021), i.e. $z_i = h_{fe}(x_i)$, which follows previous works of error slice discovery (Eyuboglu et al., 2022; Wang et al., 2023b). Then we construct a kNN graph $G = (V, E, Q)$ based on the embeddings $\{z_i\}_{i=1}^n$, which is a widely adopted manifold learning approach (Zemel & Carreira-Perpiñán, 2004; Pedronette et al., 2018; Dann et al., 2022). In the graph $G$, the edge weight $q_{ij} = 1$ if $z_j$ is among the $k$ nearest neighbors of $z_i$, or else $q_{ij} = 0$.

For the convenience of optimization, instead of hard selection, we assign a sample weight $w_i$ for each data point $(x_i, y_i)$, which is the variable to be optimized and is restricted in the range $[0, 1]$. Considering the model risk, we employ the weighted average mean of loss $\sum_{i=1}^n w_i l_i$ as our optimization objective, where $l_i = \ell(f_{\theta_0}(x_i), y_i)$ is the model prediction loss of $i$th sample given $f_{\theta_0}$. Considering coherence, we adopt manifold compactness in Definition 1 as the optimization objective, i.e. $\sum_{i=1}^n \sum_{j=1}^n w_i w_j q_{ij}$. We add these two objectives together along with a hyperparameter $\lambda$. Besides, we restrict the size of the identified slice to be no more than a proportion $\alpha$ of the dataset. Thus we formulate the optimization problem as a quadratic programming (QP) problem below:

$$\max_{\{w_i\}_{i=1}^n} \sum_{i=1}^n w_i l_i + \lambda \sum_{i=1}^n \sum_{j=1}^n w_i w_j q_{ij}$$

$$s.t. \sum_{i=1}^n w_i \leq \alpha n \tag{2}$$

$$0 \leq w_i \leq 1, \quad \forall 1 \leq i \leq n$$

The above QP problem can be easily solved by classic optimization algorithms or powerful mathematical optimization solvers like Gurobi (Gurobi Optimization, 2021). After solving for the proper sample weights $\{w_i\}_{i=1}^n$, we select the top $\alpha n$ samples sorted by the weights as the error slice $\hat{\mathcal{S}}$. Note that in most previous algorithms' workflow, they require the prediction probability as the input (Eyuboglu et al., 2022; Plumb et al., 2023; Wang et al., 2023b), thus only applicable to classification, while our algorithm takes the prediction loss as input, naturally more flexible and applicable to various tasks.

## 5 EXPERIMENTS

In this section, we conduct extensive experiments to demonstrate the validity of our proposed metric and the advantages of our algorithm MCSD compared with previous methods. For quantitative results, we conduct experiments on the error slice discovery benchmark *dcbench* (Eyuboglu et al., 2022). Besides, we conduct experiments on other types of datasets like classification for medical images (Irvin et al., 2019), object detection for driving (Yu et al., 2020), and detection of toxic comments (Borkan et al., 2019), which showcase the great potential of our algorithm to be applied to various tasks. Before we start, we briefly list the baselines:

- Spotlight (d'Eon et al., 2022): It learns a point in the embedding space as the risky centroid, and chooses the closest points to the centroid as the error slice.
- Domino (Hendrycks & Gimpel, 2016): It develops an error-aware Gaussian mixture model (GMM) by incorporating predictions into the modeling process of GMM.
- PlaneSpot (Plumb et al., 2023): It combines the prediction confidence and the reduced two-dimensional representation together as the input of a GMM.

Note that in all our experiments, we apply algorithms to the validation dataset $\{x_i^{\text{va}}, y_i^{\text{va}}\}_{i=1}^{n_{\text{va}}}$ to obtain the slicing function $g_\varphi$, and then employ $g_\varphi$ on the test dataset $\{x_i^{\text{te}}, y_i^{\text{te}}\}_{i=1}^{n_{\text{te}}}$ to acquire the prediction probability of each test sample belonging to the error slice. We choose the top $\alpha n_{\text{te}}$ samples from the test dataset sorted by the prediction probabilities as the error slice $\hat{\mathcal{S}}$, and calculate evaluation metrics based on it. As for our method that outputs the error slice of the validation dataset instead of a slicing function, to compare with other methods, we additionally train a binary MLP classifier on top of embeddings, i.e. $g_\varphi : \mathcal{Z} \mapsto [0, 1]$, by treating samples in the error slice as positive examples, and treat the rest as negative ones. However, it is worth noting that our method is effective at error slice discovery without this additional slicing function, which is illustrated in Appendix A.11.

Besides, following previous works of error slice discovery (Eyuboglu et al., 2022; Wang et al., 2023b), for image data, we employ the image encoder of CLIP with a backbone of ViT-B/32 to extract embeddings of images for error slice discovery algorithms in our main experiments. For text data, we employ pretrained BERT$_{\text{base}}$ to extract embeddings. We conduct additional experiments to show that our method is flexible in the choice of the feature extractor $h_{fe}$, whose detailed results are left in Appendix A.3. Due to the space limit, for all case studies of visual tasks, we only exhibit 3 or 5 images randomly sampled from each identified slice of the test dataset, and for baselines we only exhibit images from the slice identified by Domino, the previous SOTA algorithm. We put more examples including those of Spotlight and PlaneSpot in Appendix A.4, about 20 images for each identified slice. For running time comparison and related analyses of our method and the baselines, we leave results in Appendix A.5. For the choice and analyses of hyperparameters, we leave them in Appendix A.6. For the improvement of the original models utilizing the discovered error slices, we leave results in Appendix A.7.

In addition to coherence, we also compute the average performance of the given model $f_{\theta_0}$ on the identified slice $\hat{\mathcal{S}}$. For classification tasks, the performance metric is average accuracy. For object detection, it could be Average Precision (AP). Note that now there are two evaluation metrics at the same time. In this case, we put more emphasis on coherence instead of performance, since we only require the performance of the identified slice to be low to a certain degree but expect it to be as coherent as possible for the benefits of interpretation. This is similar to dcbench (Eyuboglu et al., 2022) where coherence also outweighs performance and is chosen as the main evaluation metric.

### 5.1 BENCHMARK RESULTS: DCBENCH

Dcbench (Eyuboglu et al., 2022) offers 886 publicly available settings for error slice discovery. Each setting consists of a trained ResNet-18 (He et al., 2016), a validation dataset and a test dataset, both

Table 1: Results of dcbench. We mark the best method in bold type and underline the second-best method in terms of each metric. "Comp." means "Manifold Compactness". "Corr." means "Correlation". "↑" indicates that higher is better. "%" indicates that the digits are percentage values.

| Metric | Precision@10 (%) ↑ | | | Precision@25 (%) ↑ | | | Average Precision (%) ↑ | | | Manifold Comp.↑ | | |
|---|---|---|---|---|---|---|---|---|---|---|---|---|
| Method | Corr. | Rare | Noisy | Corr. | Rare | Noisy | Corr. | Rare | Noisy | Corr. | Rare | Noisy |
| Spotlight | 32.3 | 28.7 | 43.2 | 32.2 | 26.4 | 40.9 | 28.9 | 16.4 | 22.7 | 4.78 | 2.67 | 4.20 |
| Domino | 36.2 | 52.5 | 51.7 | 33.8 | 52.3 | 50.0 | 29.9 | 37.7 | 31.3 | 4.14 | 4.06 | 5.53 |
| PlaneSpot | 26.1 | 18.1 | 29.4 | 22.3 | 18.1 | 27.8 | 21.8 | 14.3 | 18.8 | 2.93 | 1.59 | 3.30 |
| MCSD | **47.4** | **61.1** | **60.6** | **45.6** | **59.8** | **57.4** | **40.3** | **52.4** | **38.4** | **6.22** | **7.81** | **8.71** |

with predefined underperforming slice labels. The validation dataset and its error slice labels are taken as input of slice discovery methods, while the test dataset and its error slice labels are used for evaluation. There are three types of slices in dcbench: correlation slices, rare slices, and noisy label slices. The correlation slices are generated from CelebA (Liu et al., 2015), while the other two types of slices are generated from ImageNet (Deng et al., 2009). More details are included in Appendix A.8. In terms of evaluation metrics, we employ precision@$k$ and average precision following dcbench's practice, where precision@$k$ is the proportion of samples with top $k$ highest probabilities output by the learned slicing function that belongs to the predefined underperforming slice, and average precision is calculated based on precision@$k$ with different values of $k$. We also calculate manifold compactness as Definition 1. For all these metrics, a higher value indicates higher coherence of the identified slice, thus implying a more effective algorithm capable of error slice discovery.

**Effectiveness of our method** Table 1 shows that MCSD outperforms other methods across all three types of error slices in precision@10, precision@25, average precision, and manifold compactness. This greatly exhibits the strengths of our method compared with existing ones in error slice discovery. Among the baselines, Domino consistently ranks 2nd, also showing a fair performance.

**Validity of our metric** It is also worth noting that the proposed metric manifold compactness shows a strong consistency with other metrics. In Table 1, we find that the rank order of the four methods based on precision metrics is always MCSD, Domino, Spotlight, PlaneSpot, which is generally the same as the rank order based on manifold compactness, except for the correlation slice where the rank order of Domino and Spotlight switches. While other metrics require access to labels of predefined underperforming slices, our metric does not rely on them. This demonstrates the validity and advantages of our proposed manifold compactness when measuring coherence and evaluating the error slice discovery algorithms.

## 5.2 CASE STUDY: CELEBA

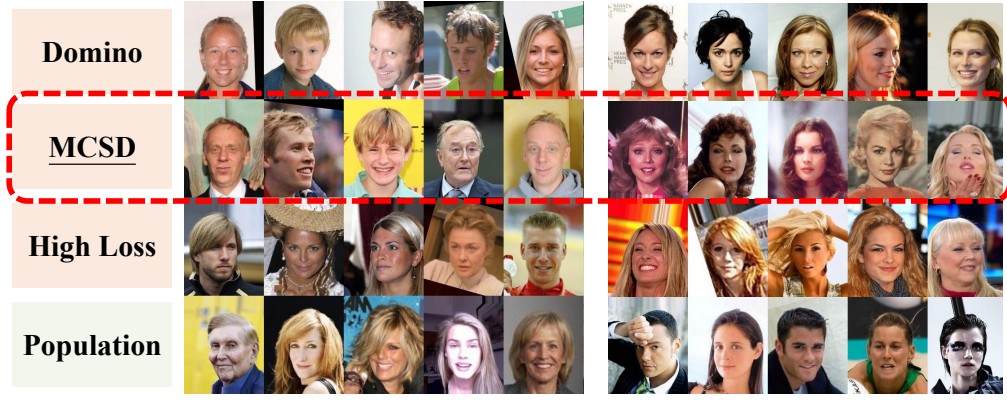

Figure 4: Images randomly sampled from slices of CelebA. Left five columns are results of the category "Blond Hair". Right five columns are results of the category "Not Blond Hair". We can see that MCSD is capable of finding error slices that are more coherent than others.

CelebA (Liu et al., 2015) is a large facial dataset of 202,599 images, each with annotations of 40 binary attributes. In the setting of subpopulation shift, it is the most widely adopted dataset since

it is easy to generate spurious correlations between two specific attributes by downsampling the dataset (Yang et al., 2023; Sagawa et al., 2019; Liu et al., 2021). Different from settings in dcbench, in this case study we follow (Sagawa et al., 2019) to treat the binary label of blond hair as the target of prediction and directly use the whole dataset of CelebA (Liu et al., 2015) without downsampling, thus closer to the real scenario. In terms of implementation details, we employ the default data split provided by CelebA and follow the training process of ERM in (Sagawa et al., 2019) to train a ResNet-50. We apply error slice discovery algorithms on both categories respectively, thus taking advantage of outcome labels that are known during slice discovery. We also illustrate results of directly selecting top $\alpha n_{te}$ samples sorted by prediction losses.

From Table 2, we can see that for both categories of CelebA, our algorithm identifies the most coherent underperforming slice in terms of manifold compactness, where higher is better. Although it ranks 2nd for the category of blond hair in terms of accuracy, where lower is better, for the task of error slice discovery, we put more emphasis on coherence since we want the identified slices to be interpretable, and we only require the performance of the slice to be lower than a threshold compared with the overall performance, as stated in Section 3. In Figure 4, the left five columns and the right five columns are from the two categories separately. The four rows correspond to randomly sampled images from different sources: the error slice that Domino identifies, the error slice that MCSD identifies, the top $\alpha n_{te}$ samples sorted by the loss, and all samples of the corresponding category. We can see that the images from the error slice identified by MCSD obviously exhibit more coherent characteristics than others.

For the category of blond hair, images in the row of MCSD are all faces of males, which conforms to the intuition that models may learn the spurious correlation between blond hair and female, and could be inclined to make mistakes in subgroups like males with blond hair in the row of MCSD. Although more than half of the images for Domino in the blond hair category are also males, its coherence is much smaller than that of MCSD, making it hard for humans to interpret the failure pattern when compared with images of the whole population. Besides, in the third row, when simply taking account of the prediction loss to select risky samples, it is also difficult to extract the common pattern. For the category of not blond hair, although both

Table 2: Results on CelebA, along with the overall accuracy of the trained model. "Acc." means "Accuracy". "Comp." means "Manifold Compactness". "↑" indicates that higher is better, while "↓" indicates that lower is better. We mark the best method in bold type and underline the second-best. "%" indicates that the digits are percentage values.

| Blond Hair? | Yes | | No | |
|---|---|---|---|---|
| Method | Acc. (%)↓ | Comp.↑ | Acc. (%)↓ | Comp.↑ |
| Spotlight | **26.3** | 5.71 | **65.9** | 3.35 |
| Domino | 34.6 | 6.07 | 82.1 | 3.58 |
| PlaneSpot | 68.4 | 2.92 | 93.6 | 1.13 |
| MCSD | 33.8 | **8.09** | 75.7 | **5.54** |
| Overall | 76.4 | - | 98.2 | - |

Domino and sorting-by-loss can extract the pattern of faces being female with brown hair or blond hair (label noise), MCSD identifies more detailed common characteristics that faces in the images are not only female, but bear vintage styles like in the 20th century, which also constitute a riskier slice than Domino in terms of accuracy. It is also worth noting that MCSD achieves a higher manifold compactness than Domino in Table 2, consistent with that the identified slice of MCSD exhibits more coherent semantics in Figure 4, further confirming the rationality of our proposed coherence metric.

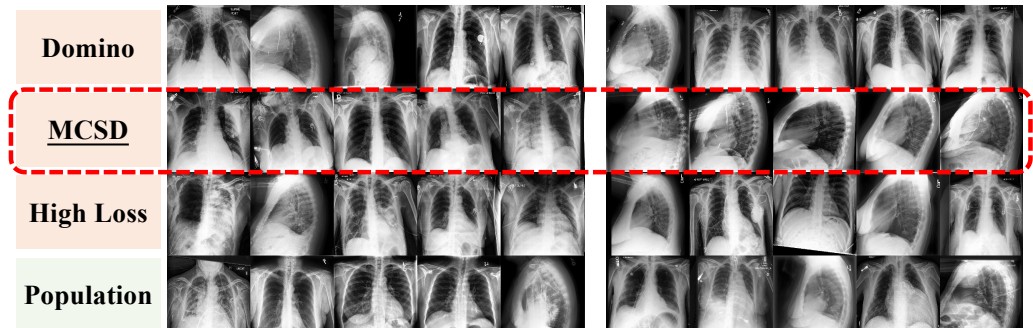

Figure 5: Images randomly sampled from slices of CheXpert. Left five columns are results of the category "Ill". Right five columns are results of the category "Healthy". We can see that MCSD is capable of finding error slices that are more coherent than others.

## 5.3 Case Study: CheXpert

To demonstrate the effectiveness of our algorithm on other types of data, we conduct experiments on a medical imaging dataset, i.e. CheXpert (Irvin et al., 2019), where the task is to predict whether patients are ill or not based on their chest X-ray images. It contains 224,316 images coming from 65,240 patients. We follow the data split and training process of (Yang et al., 2023) to train a ResNet-50. Still, we apply algorithms to images of ill and healthy patients respectively.

In Table 3, we can see that MCSD still achieves highest manifold compactness and relatively low slice accuracy in terms of the discovered error slice for both ill and healthy patients. In Figure 5, for ill patients, images sampled from the error slice discovered by MCSD are all taken from the frontal view, while there are different views for images sampled from other sources. For healthy patients, images corresponding to MCSD are all taken from the left lateral view, while other rows constitute images from different views, making it difficult to extract the common risky pattern. These results showcase MCSD's usefulness in medical imaging, which is a highly risk-sensitive task and deserves more attention for error slice discovery and failure pattern interpretation. Besides, the consistency of the order of coherence for MCSD and Domino in Table 3 and Figure 5 also confirms the rationality of our proposed coherence metric.

Table 3: Results on CheXpert, along with the overall accuracy of the trained model. "Acc." means "Accuracy". "Comp." means "Manifold Compactness". "↑" indicates that higher is better, while "↓" indicates that lower is better. We mark the best method in bold type and underline the second-best. "%" indicates that the digits are percentage values.

| Ill? | Yes | | No | |
|---|---|---|---|---|
| Method | Acc. (%) ↓ | Comp.↑ | Acc. (%) ↓ | Comp.↑ |
| Spotlight | **19.5** | 2.10 | 64.9 | 4.70 |
| Domino | 31.5 | 1.53 | 88.4 | 2.82 |
| PlaneSpot | 42.8 | 3.66 | 69.5 | 3.17 |
| MCSD | 31.5 | **4.70** | **63.3** | **4.87** |
| Overall | 45.5 | - | 91.0 | - |

## 5.4 Case Study: BDD100K

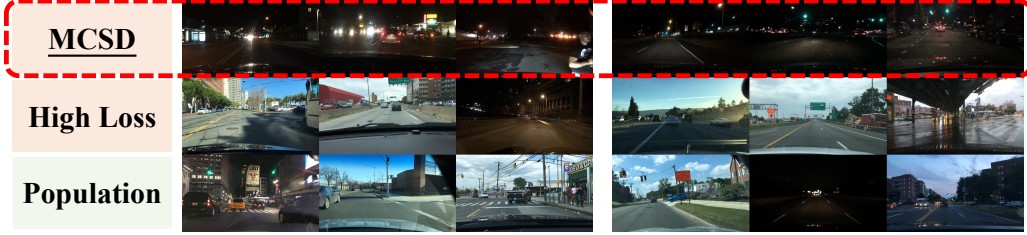

Figure 6: Images randomly sampled from slices of BDD100K. Left three columns are results of the category "Pedestrian". Right three columns are results of the category "Traffic Light". We can see that MCSD is capable of finding error slices that are more coherent than others.

Compared with most previous algorithms (Eyuboglu et al., 2022; Wang et al., 2023b; Plumb et al., 2023) that require prediction probabilities as a part of input and are only designed for classification tasks, our algorithm MCSD is flexible to be employed in various tasks since it takes prediction losses as input. To illustrate its benefits of extending to other tasks, we conduct a case study on BDD100K (Yu et al., 2020), a large-scale dataset composed of driving scenes with abundant annotations. It includes ten tasks, of which we investigate object detection in our paper. The number of images in BDD100K's object detection task is 79,863, which we split into train, validation,

Table 4: Results of algorithms on BDD100K for two categories, along with the overall AP of the trained model. "Comp." means "Manifold Compactness". "↑" indicates that higher is better, while "↓" indicates that lower is better. We mark the best method in bold type. "%" indicates that the digits are percentage values.

| Category | Pedestrian | | Traffic Light | |
|---|---|---|---|---|
| Method | AP (%) ↓ | Comp.↑ | AP (%) ↓ | Comp.↑ |
| Spotlight | 57.3 | 2.05 | **46.3** | 2.61 |
| MCSD | **53.8** | **6.60** | 57.3 | **4.78** |
| Overall | 71.4 | - | 69.2 | - |

and test datasets with the ratio 2:1:1. We train a YOLOv7 (Wang et al., 2023a) and try to identify coherent error slices for it. We employ Average Precision (AP) as the metric of performance that is widely adopted in detection tasks. Of the 13 categories in the task, we select 2 categories with a relatively high overall performance and a large sample size, i.e. pedestrian and traffic light. We apply our algorithm MCSD for each of them respectively. Note that we could not compare with Domino or PlaneSpot since neither of them is applicable to tasks other than classification.

In Table 4, we can see that MCSD successfully identifies error slices whose AP are lower than those of overall for both categories, and whose coherence is higher than that of Spotlight in terms of manifold compactness. In Figure 6, each row corresponds to five images randomly sampled from a given source. The left three columns correspond to the category of pedestrians, while the right three columns correspond to the category of traffic lights. For both pedestrians and traffic lights, samples from the source of MCSD are coherent in that they are all taken at night. This conforms to the intuition that it is more difficult to recognize and locate objects when the light is poor. However, directly sampling from the high-loss images can hardly exhibit any common patterns. This reveals the potential of our algorithm to be extended to other types of tasks.

## 5.5 CASE STUDY: CIVILCOMMENTS

In addition to experiments on visual tasks, to demonstrate the applicability of our method to other types of data, we conduct experiments on CivilComments (Borkan et al., 2019), a text dataset included in some popular distribution shift benchmarks (Yang et al., 2023; Koh et al., 2021). Its task is to predict whether a given comment is toxic or not. We employ the version of the dataset in Yang et al. (2023) where the dataset has 244,436 comments, and follow its data split and training process to train a $BERT_{base}$. We apply algorithms to toxic and non-toxic comments respectively. In Table 5, we can see that MCSD identifies slices of the lowest accuracy and highest manifold compactness in both categories.

We also list two parts of comments that are respectively sampled from the slice identified by applying MCSD to the "toxic" category and from all comments of "toxic" category in Appendix A.9 (**Warning**: many of these comments are severely offensive or sensitive), where each part contains 10 comments. We employ Chat-GPT to tell the main difference between the two parts of comments and the reply is "Part 1 is characterized by detailed, historical, and ethical discussions with a critical stance on conservatism and a defense of marginalized groups". We further check and confirm that comments in part 1, i.e. the slice identified by our method, mostly present a positive attitude towards minority groups in terms of gender, race, or religion. This implies that the model tends to treat comments with excessively positive attitudes towards minority groups as non-toxic, some of which are actually offensive and toxic. These results demonstrate our method's usefulness in text data.

Table 5: Results on CivilComments, along with the overall accuracy of the trained model. "Comp." means "Manifold Compactness". "↑" indicates that higher is better, while "↓" indicates that lower is better. We mark the best method in bold type. "%" indicates that the digits are percentage values.

| Toxic? | Yes | | No | |
|---|---|---|---|---|
| Method | Acc. (%)↓ | Comp.↑ | Acc. (%)↓ | Comp.↑ |
| Spotlight | 48.6 | 5.10 | 91.0 | 7.33 |
| Domino | 56.1 | 5.98 | 87.9 | 6.55 |
| PlaneSpot | 46.3 | 1.65 | 96.5 | 2.99 |
| MCSD | **25.2** | **8.56** | **60.8** | **7.67** |
| Overall | 61.2 | - | 90.9 | - |

## 6 CONCLUSION

In this paper, inspired by the data geometry property, we propose manifold compactness as a metric of coherence given a slice, which does not rely on predefined underperforming slice labels. We conduct empirical analyses to justify the rationality of our proposed metric. With the help of explicit metrics for risk and coherence, we develop an algorithm that directly incorporates both risk and coherence into the optimization objective. We conduct experiments on a benchmark and perform case studies on various types of datasets to demonstrate the validity of our proposed metric and the superiority of our algorithm, along with the potential to be flexibly extended to different types of tasks.

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

# A APPENDIX

## A.1 MORE EXPERIMENTAL RESULTS RELATED TO MANIFOLD COMPACTNESS

In this part, we provide more experimental results that demonstrate the validity and advantages of our proposed coherence metric, i.e. manifold compactness. In Section 3 we only present results of CelebA, while here we also present results on other datasets like CheXpert and BDD100K.

### A.1.1 VISUALIZATION ANALYSES

We provide visualization results of different dimension-reduction methods: PCA, t-SNE, and UMAP, where PCA mainly preserves pairwise Euclidean distances between data points while t-SNE and UMAP are both manifold learning techniques. We employ features extracted by the image encoder of CLIP-ViT-B/32 as input of the dimension-reduction methods. Thus the original dimension (dimension of features extracted by the image encoder of CLIP-ViT-B/32) is 512 and the reduced dimension is 2 for convenience of visualization. In Figure 7 and 8, blue dots are correctly classified by the trained model and red dots are wrongly classified. In Figure 9, the color is brighter when the loss is higher. All three visualizations illustrate that t-SNE and UMAP show much clearer clustering structures than PCA, either showing a larger number of clusters or exhibiting larger margins between clusters. Such results indicate that it is better to measure coherence in the metric space of a manifold instead of using metrics directly calculated in Euclidean space.

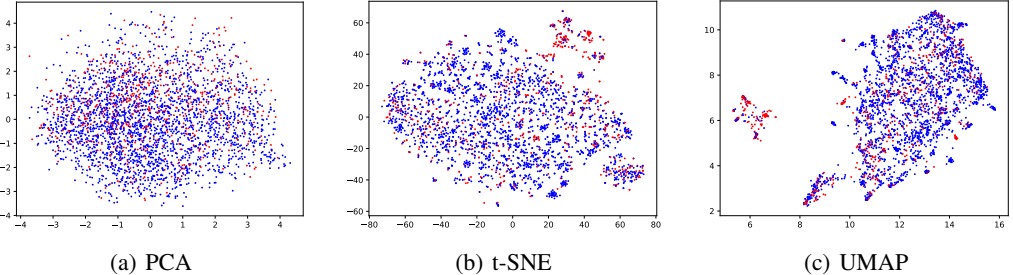

(a) PCA        (b) t-SNE        (c) UMAP

Figure 7: Visualization: Category "blond hair" of CelebA.

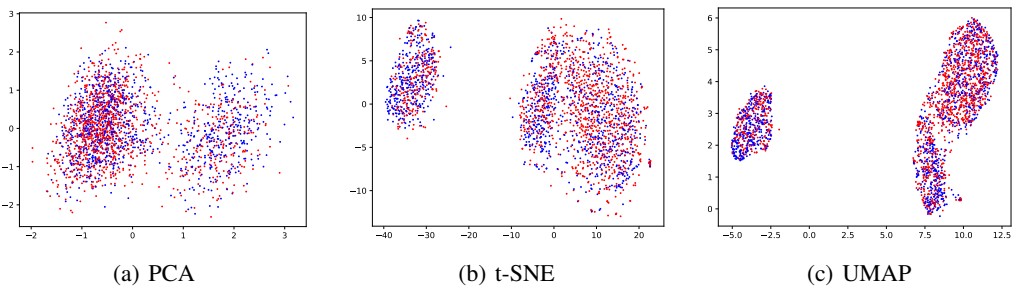

(a) PCA        (b) t-SNE        (c) UMAP

Figure 8: Visualization: Category "ill" of CheXpert.

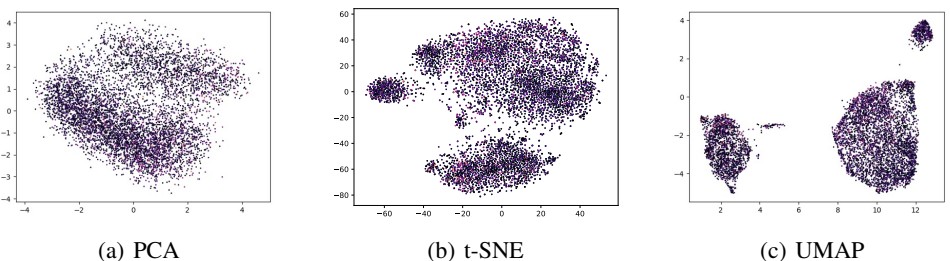

(a) PCA        (b) t-SNE        (c) UMAP

Figure 9: Visualization: Category "pedestrian" of BDD100K.

### A.1.2 COMPARISON WITH VARIANCE

We compare manifold compactness with variance, a common and representative metric of coherence directly calculated in the Euclidean space, on different semantically predefined slices. Note that since the slices are not of the same size, to compare manifold compactness of different slices properly, for each given slice we randomly sample a subset of size 150 with 20 times, and average the manifold compactness of 20 subsets as the manifold compactness of the given slice. For CelebA, we use the binary label $y$ to indicate whether the person has blond hair or not, and $a$ to indicate whether the person is male or not. From Appendix A.1.2, we can see that for manifold compactness, its value of the more fine-grained slice, i.e. the more coherent slice, is larger than the more coarse-grained slice. For example, the manifold compactness of $y = 1 \& a = 0$ is 0.38, larger than that of $y = 1$ (the value is 0.35) or $a = 0$ (the value is 0.13). Such a relationship holds for every pair of slices. However, in terms of variance, for example, variance of $y = 1 \& a = 1$ is 39.6, larger than that of $y = 1$ whose value is 36.2, which is contrary to our expectation that variance of the more fine-grained slice is smaller than that of the more coarse-grained slice. For CheXpert, we use the binary label $y$ to indicate whether the person is ill or not, and $a$ to indicate whether the person is male or not. We also find that for manifold compactness, its value of the more fine-grained slice, i.e. the more coherent slice, is larger than the more coarse-grained slice, while the value of variance is not consistent with the granularity of the slice. We also additionally compare with other metrics that are directly calculated in Euclidean distance, including Mean Absolute Deviation (MeanAD), Median Absolute Deviation (MedianAD), and Interquartile Range (IQR). We find that they exhibit similar phenomenons to variance, i.e. the metric value of the more coarse-grained slice is sometime even smaller than that of the more fine-grained slice, which contradicts our expectation. Thus we demonstrate that manifold compactness is better at capturing semantic coherence than variance does.

Table 6: Comparing manifold compactness with metrics directly calculated in Euclidean space.

| Dataset | CelebA | | | | | CheXpert | | | | |
|---|---|---|---|---|---|---|---|---|---|---|
| Slice | Comp. | Var. | MeanAD | MedianAD | IQR | Comp. | Var. | MeanAD | MedianAD | IQR |
| All | 0.07 | 42.9 | 113.9 | 96.2 | 192.9 | 0.07 | 9.4 | 42.3 | 34.7 | 69.2 |
| $y = 1$ | 0.35 | 36.2 | 102.7 | 86.5 | 173.1 | 0.12 | 10.1 | 42.1 | 34.6 | 69.3 |
| $y = 0$ | 0.08 | 43.6 | 114.6 | 97.0 | 194.3 | 0.07 | 9.4 | 42.3 | 34.6 | 69.1 |
| $a = 1$ | 0.17 | 42.5 | 114.2 | 96.4 | 193.1 | 0.08 | 9.8 | 41.3 | 33.8 | 67.5 |
| $a = 0$ | 0.13 | 38.3 | 106.9 | 90.1 | 180.4 | 0.08 | 9.0 | 43.2 | 35.4 | 70.7 |
| $y = 1, a = 1$ | 3.19 | 39.6 | 107.3 | 90.4 | 181.6 | 0.15 | 9.5 | 40.8 | 33.6 | 67.4 |
| $y = 1, a = 0$ | 0.38 | 35.2 | 101.1 | 85.4 | 171.0 | 0.17 | 10.1 | 43.1 | 35.4 | 71.2 |
| $y = 0, a = 1$ | 0.18 | 42.5 | 114.1 | 96.3 | 193.0 | 0.09 | 9.9 | 41.3 | 33.8 | 67.4 |
| $y = 0, a = 0$ | 0.15 | 38.8 | 107.1 | 90.2 | 180.7 | 0.09 | 8.6 | 43.2 | 35.4 | 70.6 |

### A.2 SHOWCASE FOR MULTIPLE ERROR SLICES

In this part, we compare both the worst slice and the second worst slice discovered by our algorithm MCSD and the previous SOTA algorithm Domino. For MCSD, we remove the first error slice from the validation dataset and apply our algorithm again to the rest of the validation dataset to acquire the second error slice. For Domino, we select the slice with the highest and second highest prediction error in the validation dataset as the worst and second worst slice. Results of the blond hair category of CelebA are shown in Figure 10. We find that MCSD is also capable of identifying a coherent slice where faces are female with vintage styles, similar to the error slice also identified by MCSD in Figure 13, while only the pattern of female can be captured in the second worst slice identified by Domino.

### A.3 CHOICE OF FEATURE EXTRACTORS

We conduct additional experiments on CelebA by changing CLIP-ViT-B/32 to CLIP-ResNet50 and ImageNet-supervised-pretrained ResNet50. Table 7 shows that whatever the pretrained feature extractor is, MCSD consistently identifies slices of low accuracy and outperforms other methods in terms of manifold compactness. It is worth noting that this conclusion is valid even for MCSD with ResNet50, which is generally considered as a weaker pretrained feature extractor than ViT-B/32 employed by baselines. In Figure 11, we can see that MCSD with different pretrained feature

Figure 10: Showcase of multiple error slices for each algorithm on the category "Blond Hair" of CelebA.

extractors truly identifies coherent error slices for the blond hair category of CelebA. As for the practice of using pretrained feature extractors, it is acceptable and generally adopted in previous

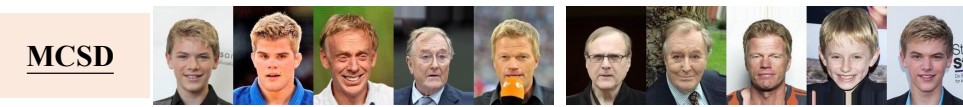

Figure 11: Left five images are sampled from the slice identified by MCSD (CLIP-ResNet50). Right five images are sampled from the slice identified by MCSD (Supervised-ResNet50).

Table 7: Experiments using different pretrained feature extractors.

| Blond Hair? | Yes | | No | |
|---|---|---|---|---|
| Method | Acc. (%)↓ | Comp.↑ | Acc. (%)↓ | Comp.↑ |
| Spotlight | **26.3** | 5.71 | **65.9** | 3.35 |
| Domino | 34.6 | 6.07 | 82.1 | 3.58 |
| PlaneSpot | 68.4 | 2.92 | 93.6 | 1.13 |
| MCSD(CLIP-ViT-B/32) | 33.8 | 8.09 | 75.7 | **5.54** |
| MCSD(CLIP-ResNet50) | 29.3 | **8.77** | 71.7 | 5.38 |
| MCSD(Supervised-ResNet50) | 32.8 | 7.22 | 67.0 | 4.75 |
| Overall | 76.4 | - | 98.2 | - |

## A.4 MORE EXAMPLES FOR CASE STUDIES

In this part, we provide more examples for the case studies of visual tasks in our main paper. For CelebA (Figure 12 and 13) and CheXpert (Figure 14 and 15), we randomly sample 20 images from each slice and put 10 images in a row. For BDD100K (from Figure 16 to 23), we randomly sample 18 images for each slice and put 3 images in a row for clearer presentation. We also draw the predicted bounding box with red color and the ground truth bounding box with yellow color. Experimental findings are basically the same as those in our main paper. MCSD still consistently identifies coherent slices in these three cases. Note that in CheXpert, previous algorithms like Spotlight and PlaneSpot are also able to identify coherent slices, illustrating a certain degree of their effectiveness in error slice discovery.

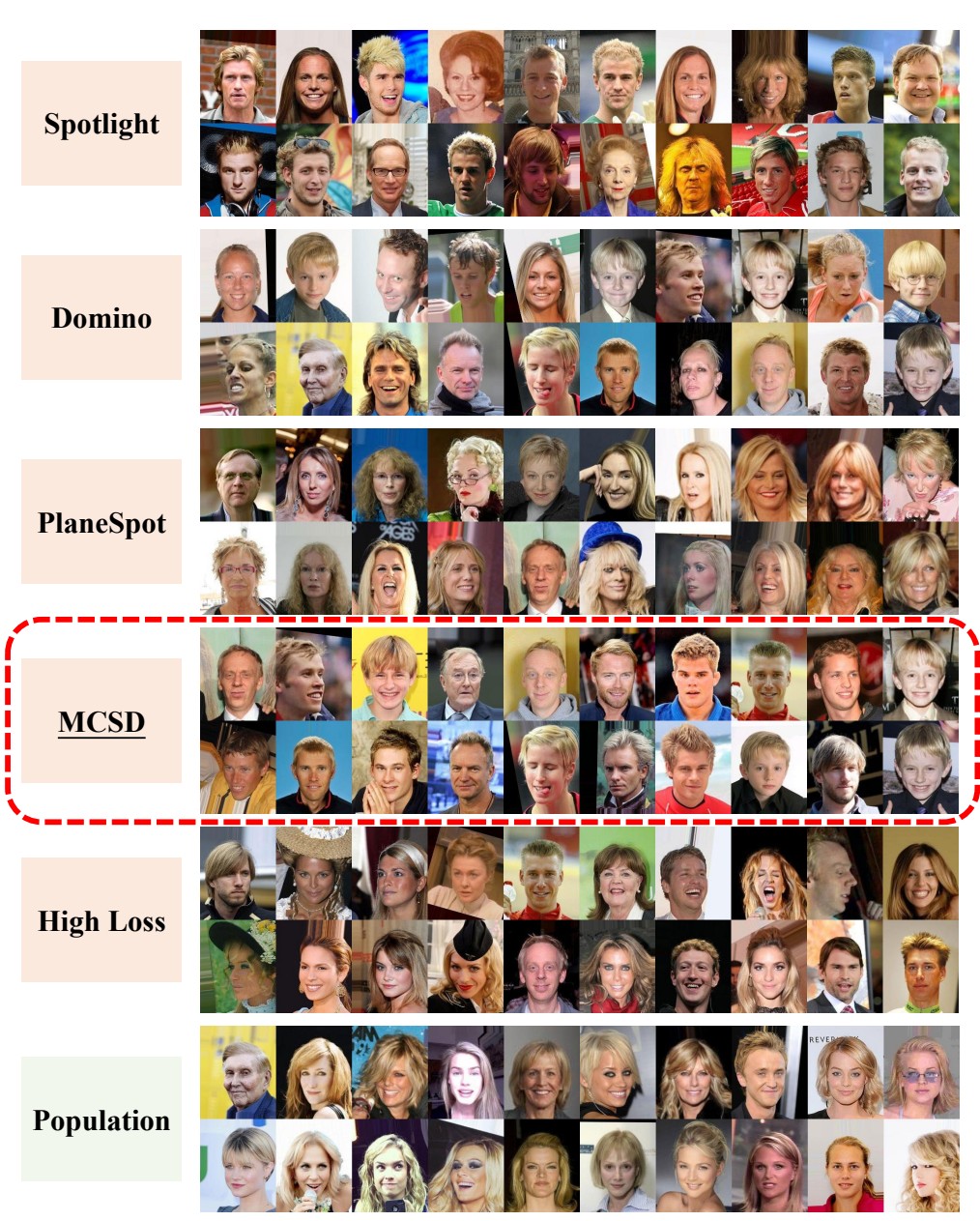

Figure 12: More examples of the category "Blond Hair" of CelebA.

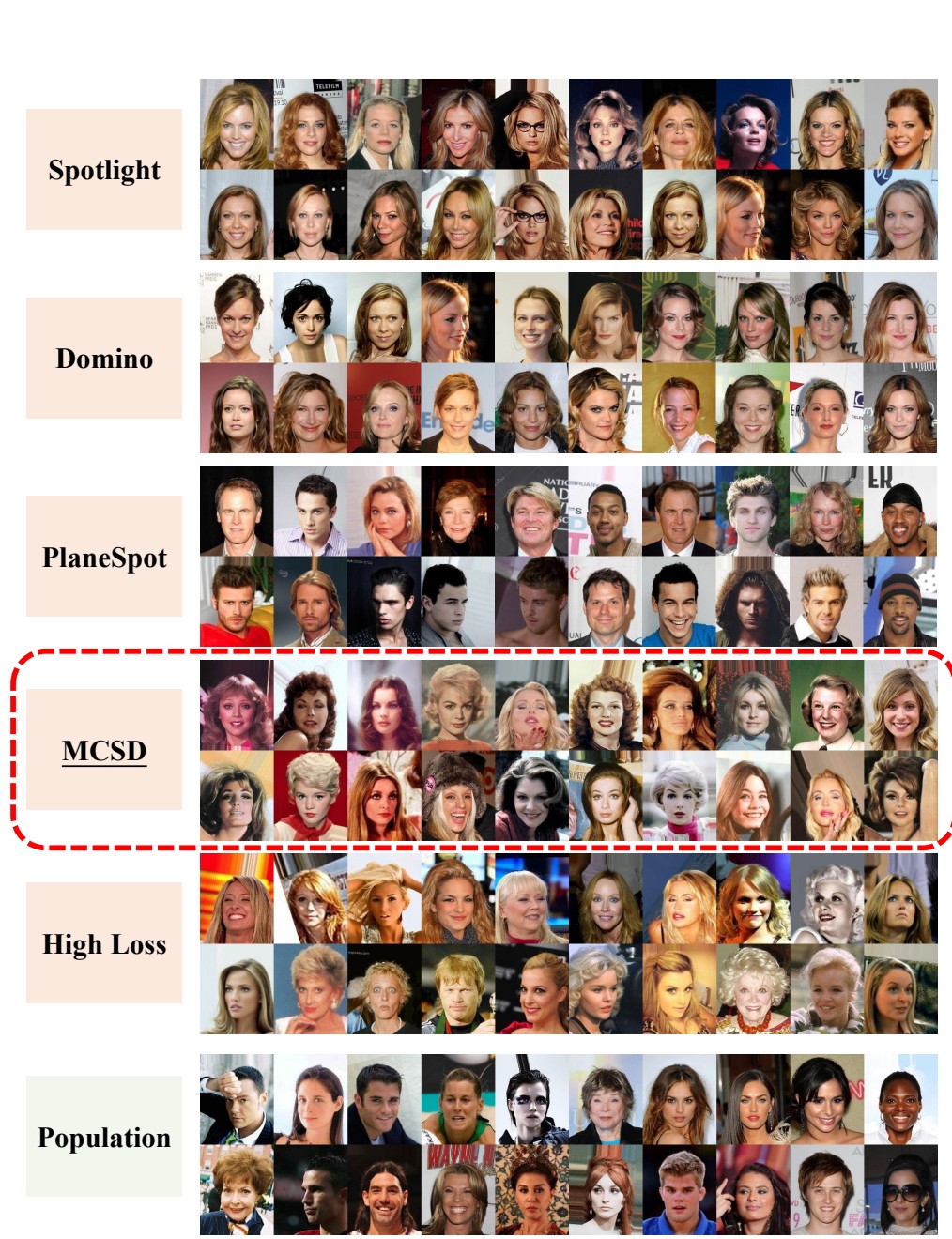

Figure 13: More examples of the category "Not Blond Hair" of CelebA.

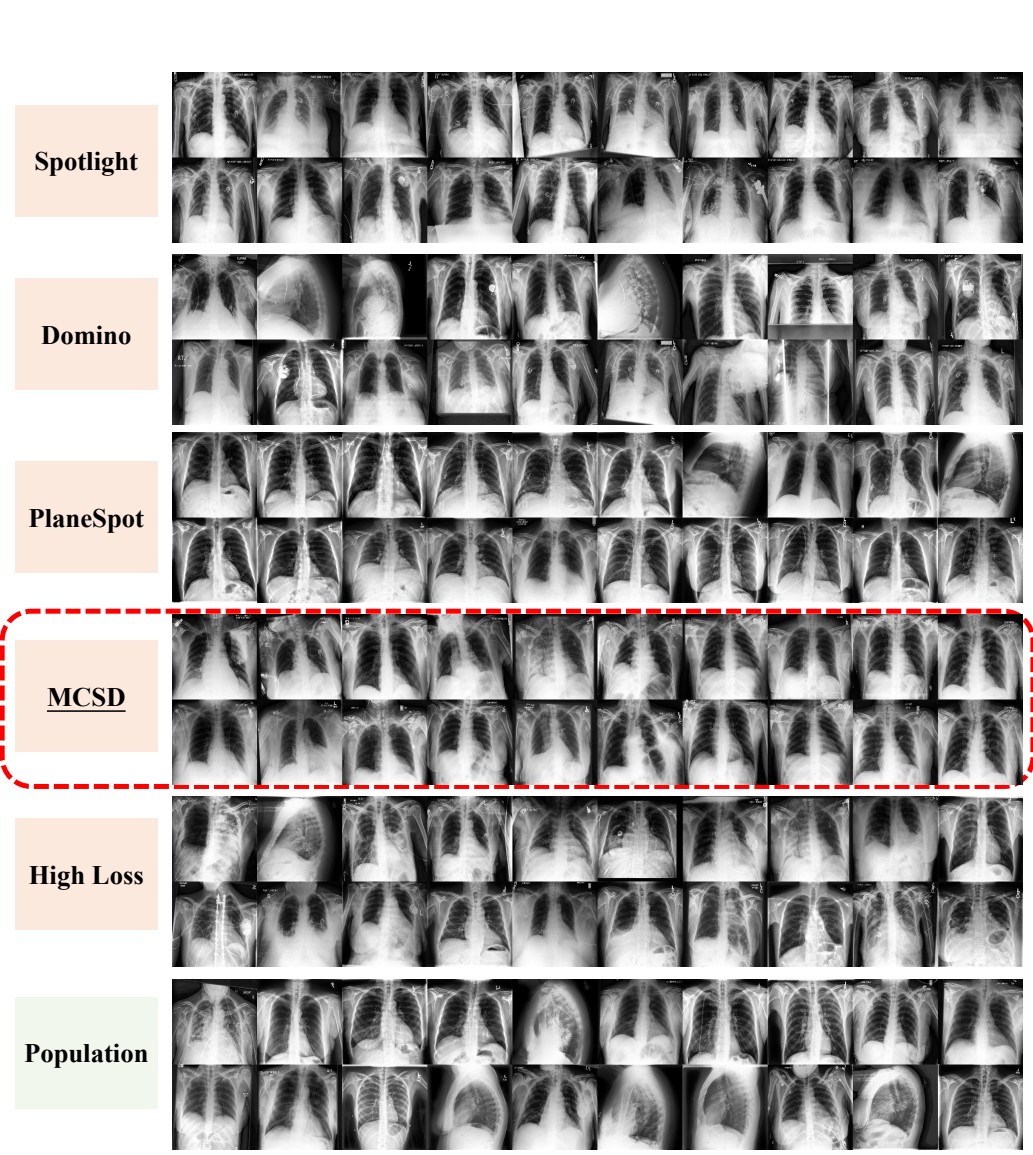

Figure 14: More examples of the category "ill" of CheXpert.

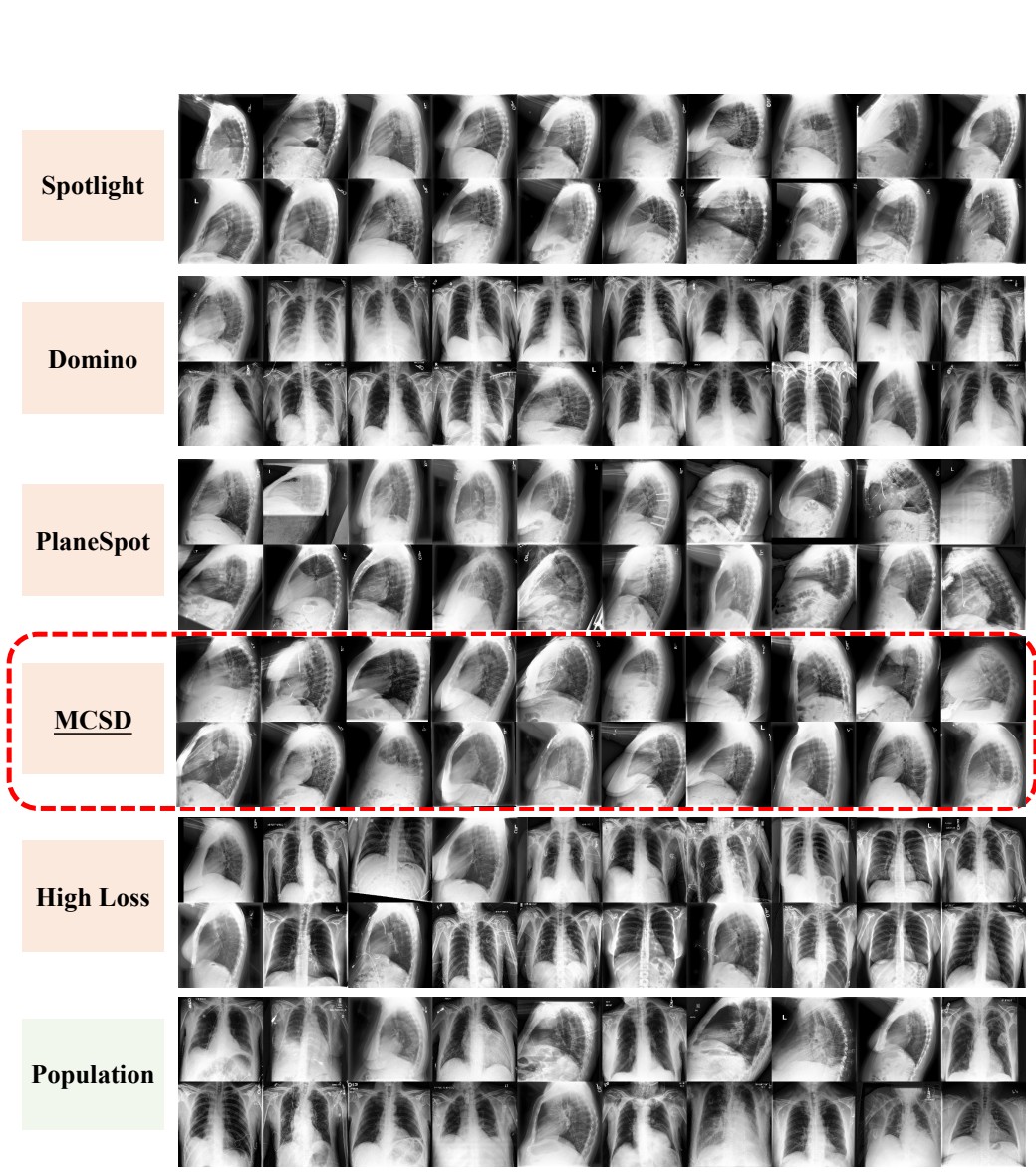

Figure 15: More examples of the category "healthy" of CheXpert.

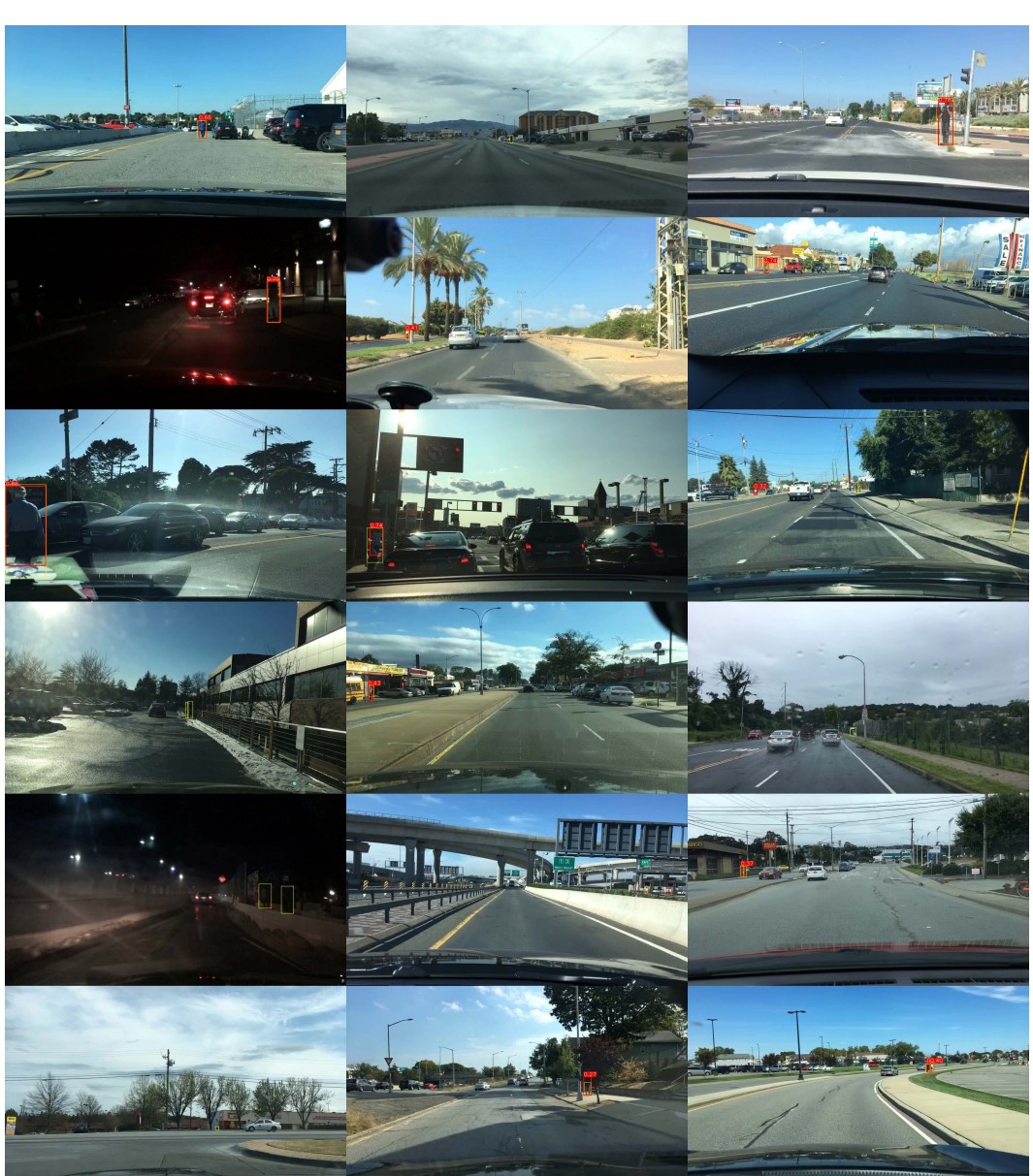

Figure 16: More examples of the category "Pedestrian" of BDD100K via Spotlight.

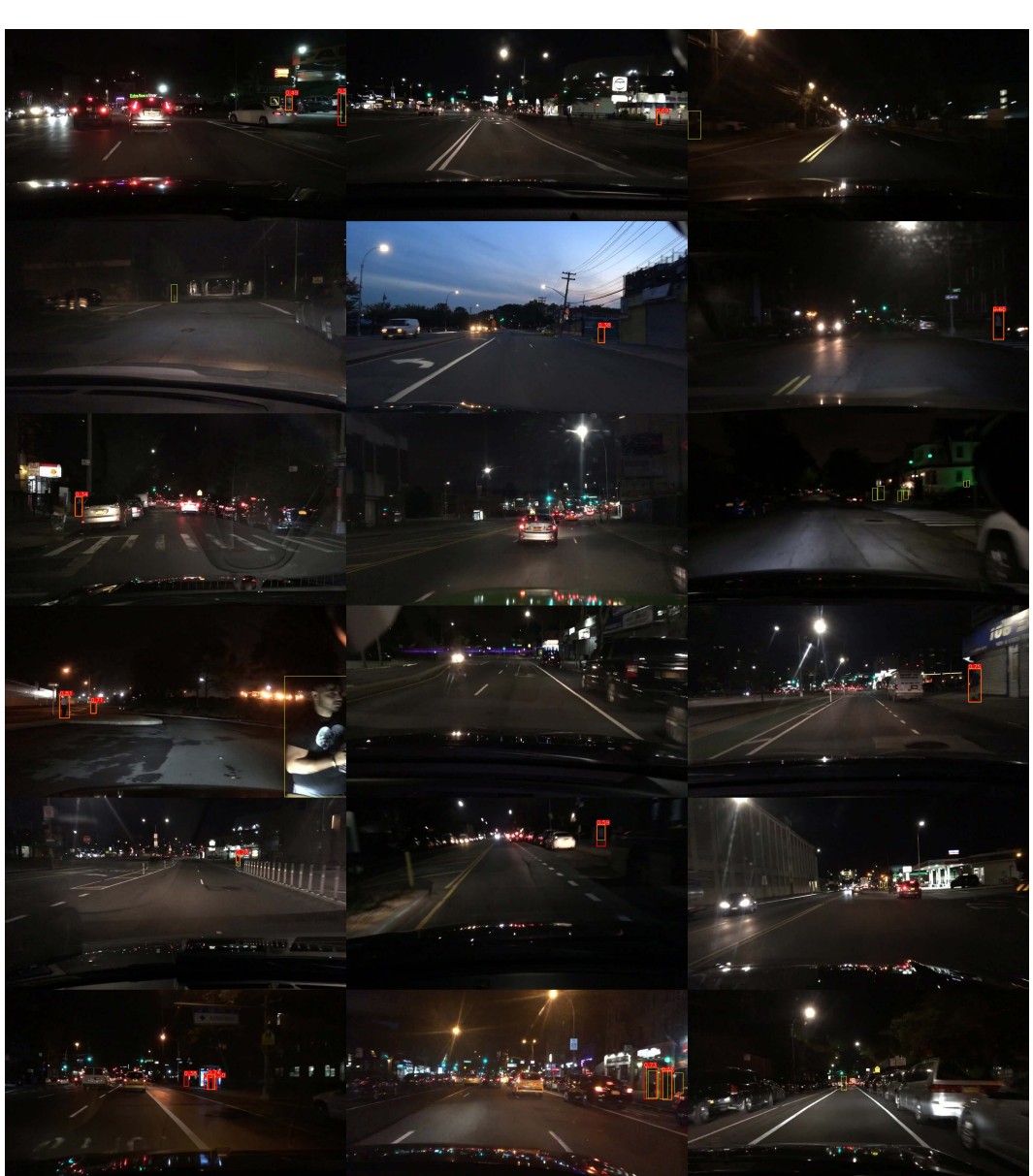

Figure 17: More examples of the category "Pedestrian" of BDD100K via MCSD.

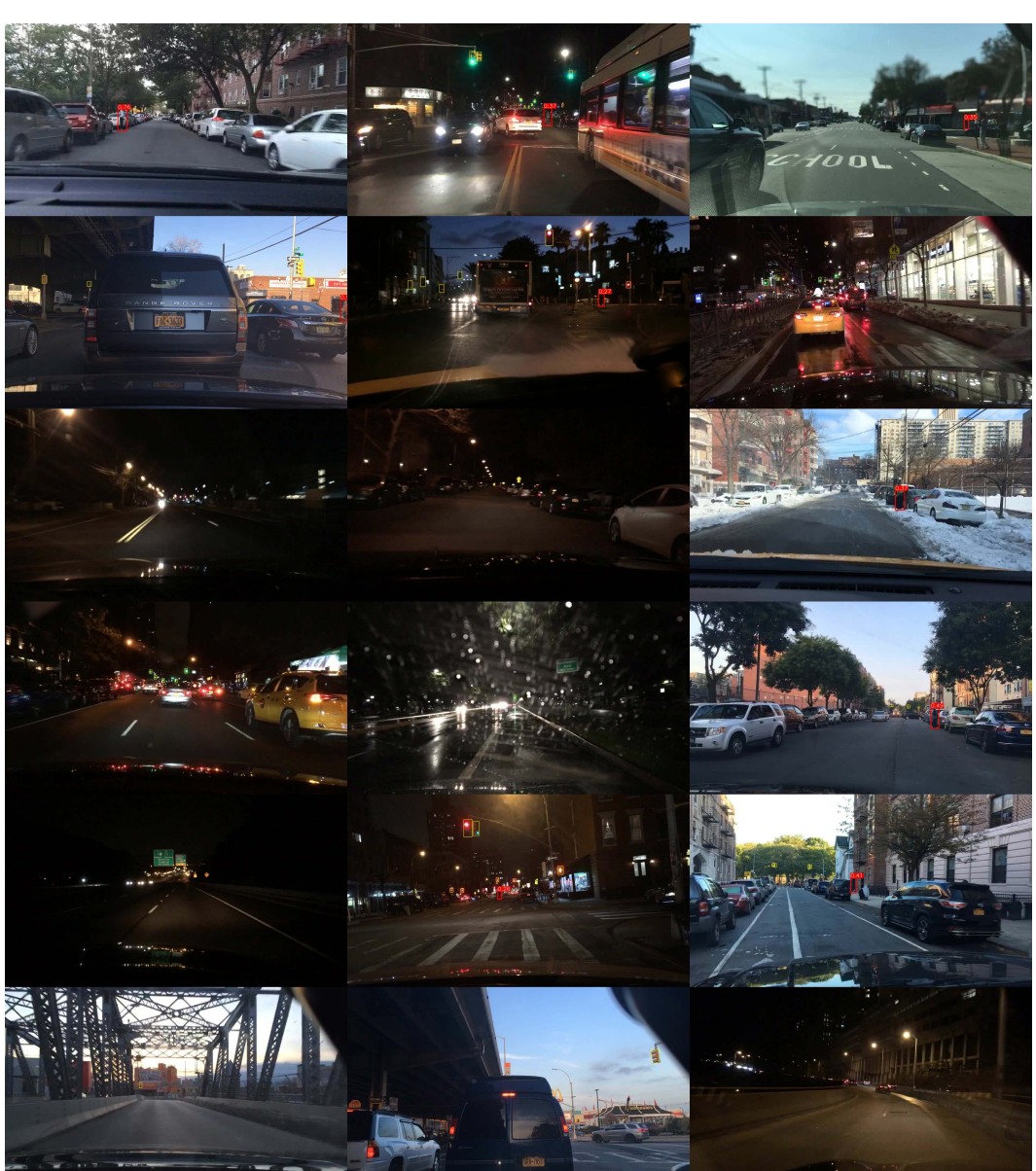

Figure 18: More examples of the category "Pedestrian" of BDD100K sampling from high loss images.

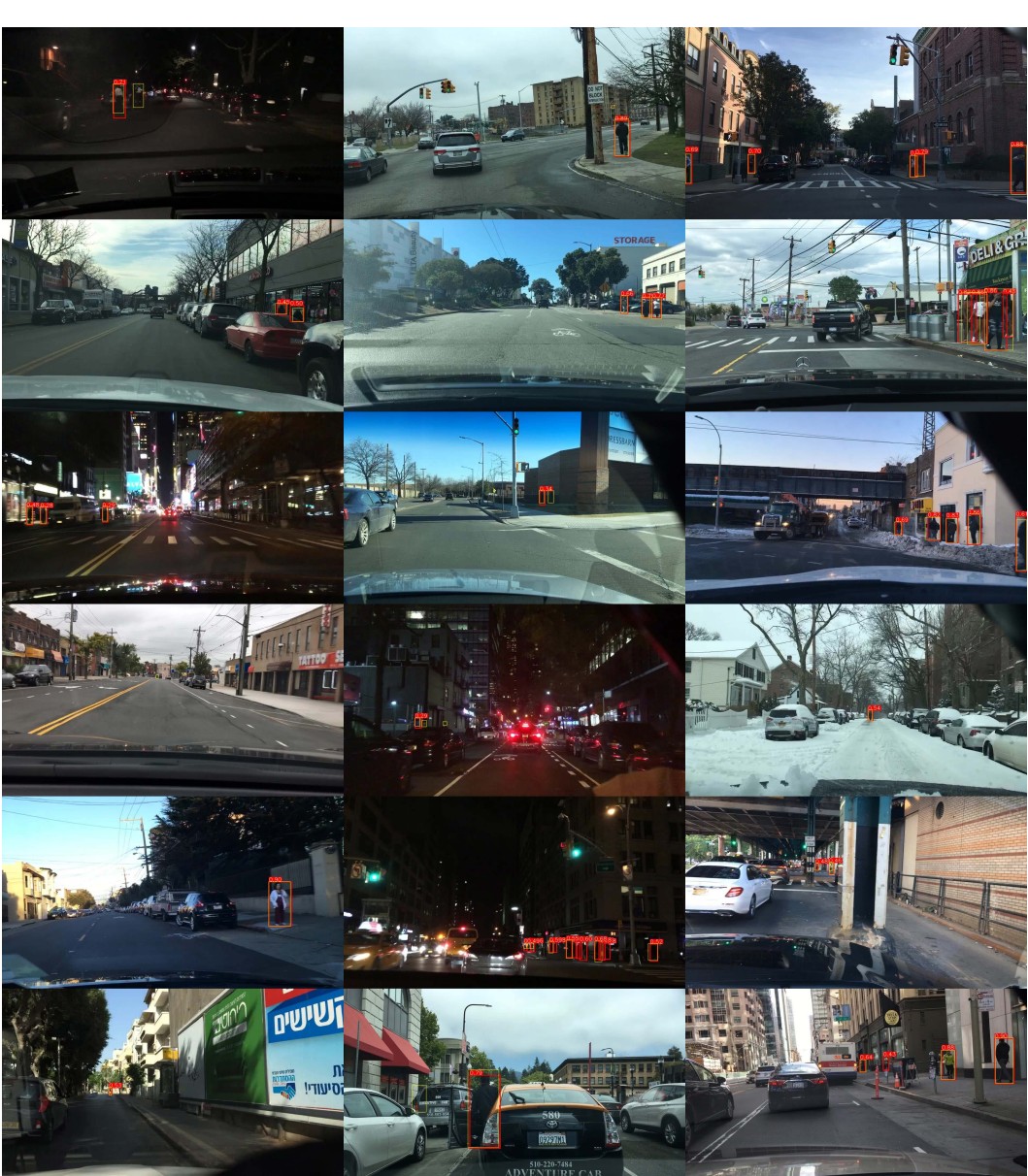

Figure 19: More examples of the category "Pedestrian" of BDD100K sampling from the whole population.

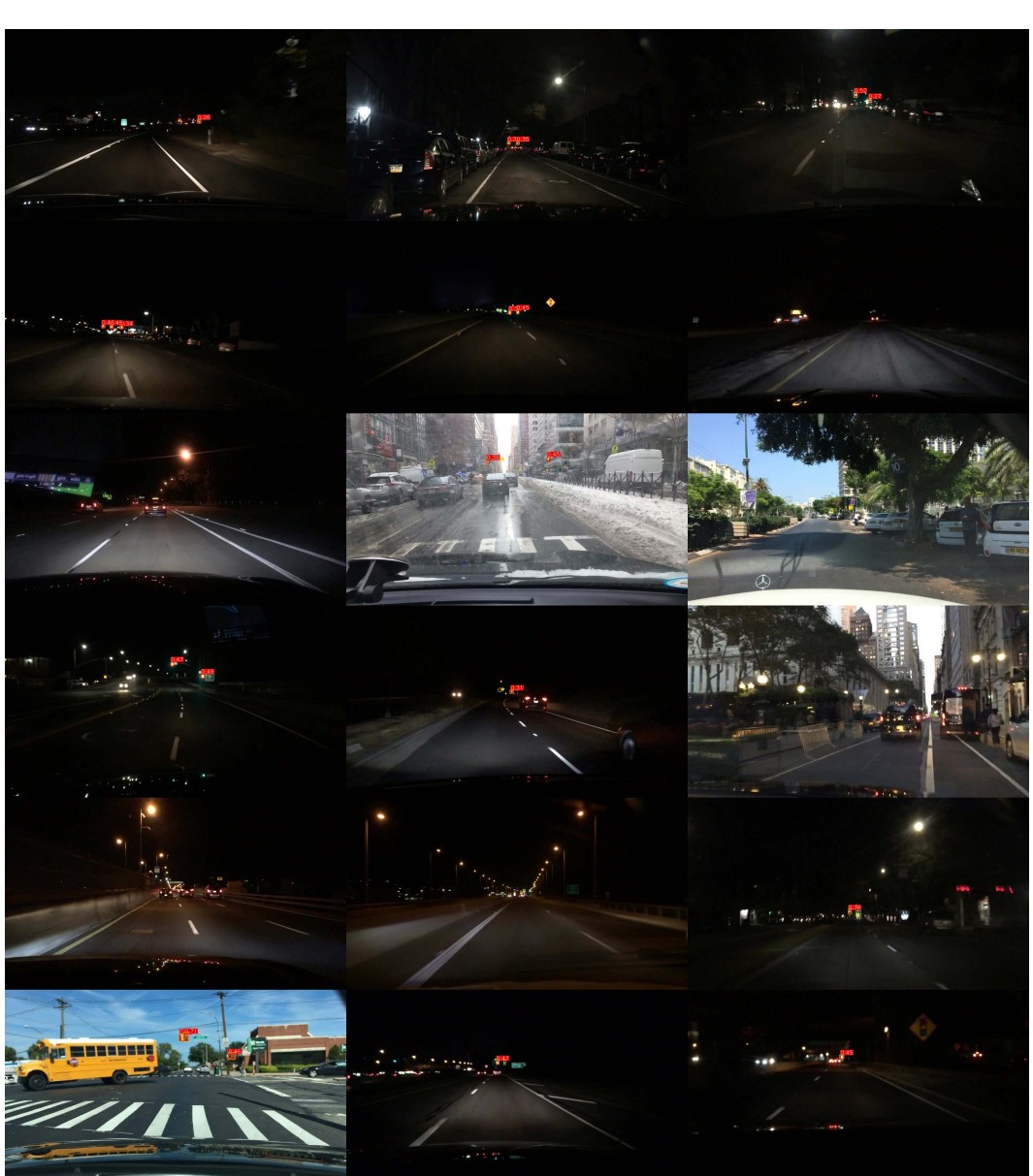

Figure 20: More examples of the category "Traffic Light" of BDD100K via Spotlight.

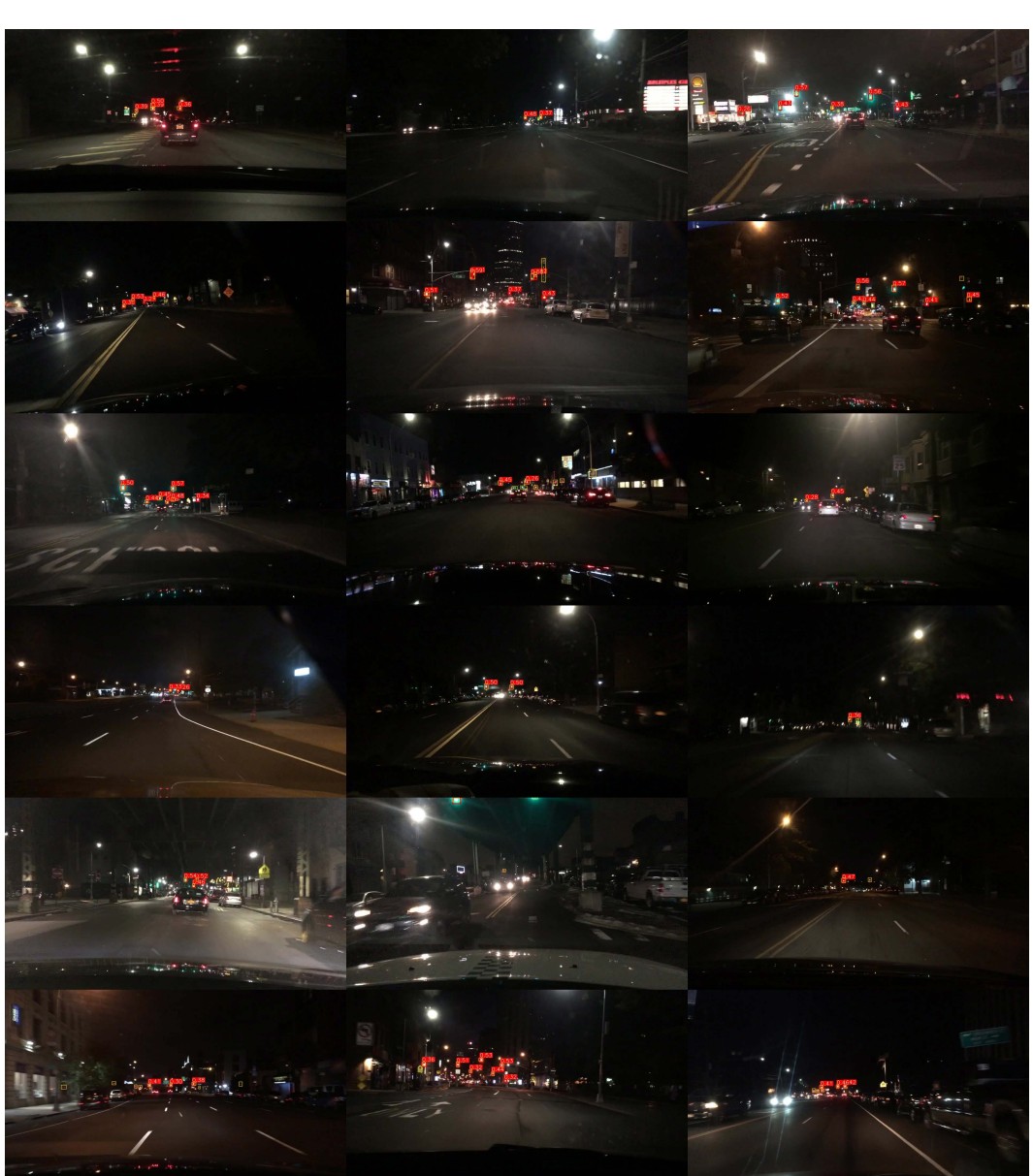

Figure 21: More examples of the category "Traffic Light" of BDD100K via MCSD.

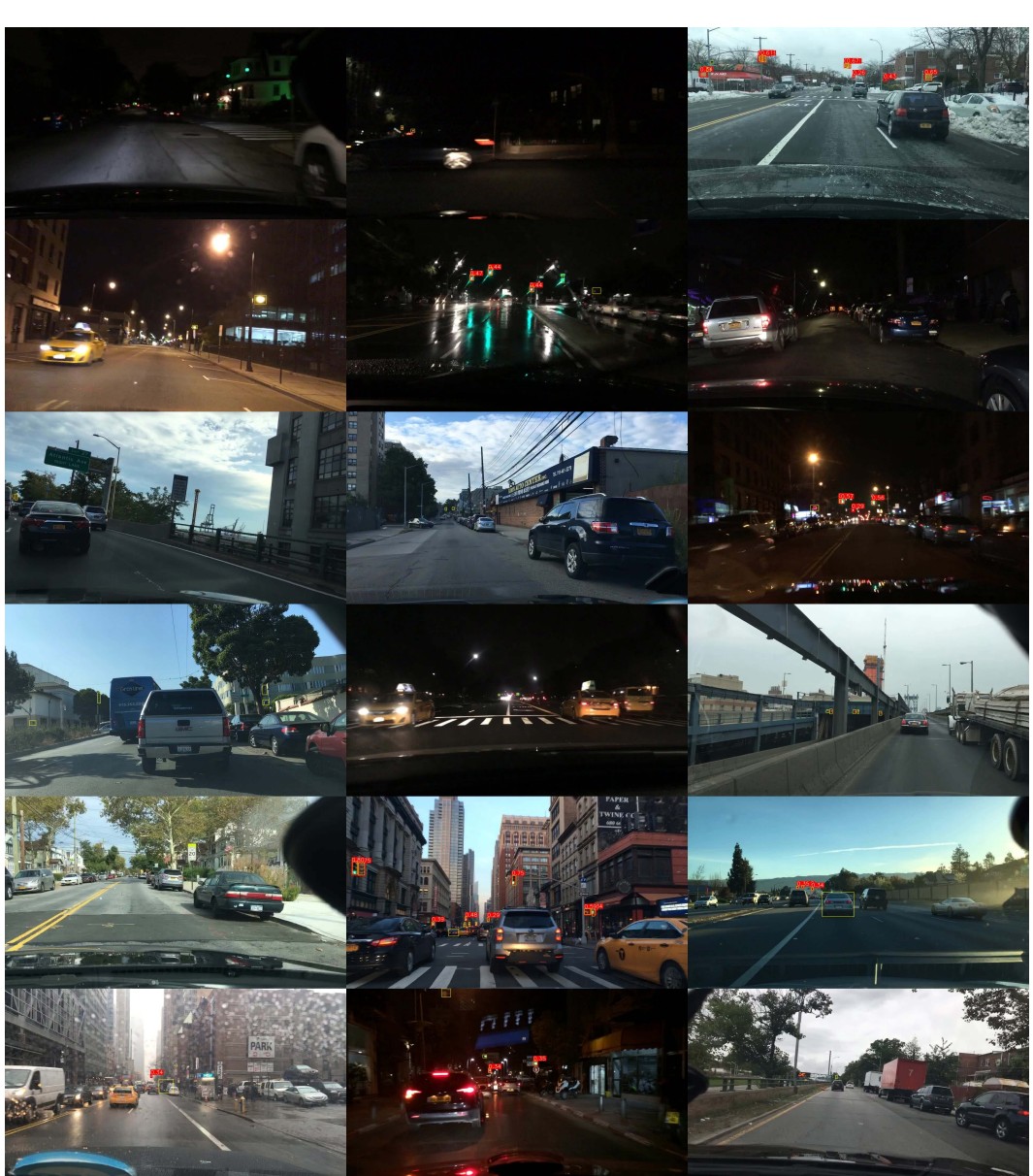

Figure 22: More examples of the category "Traffic Light" of BDD100K sampling from high loss images.

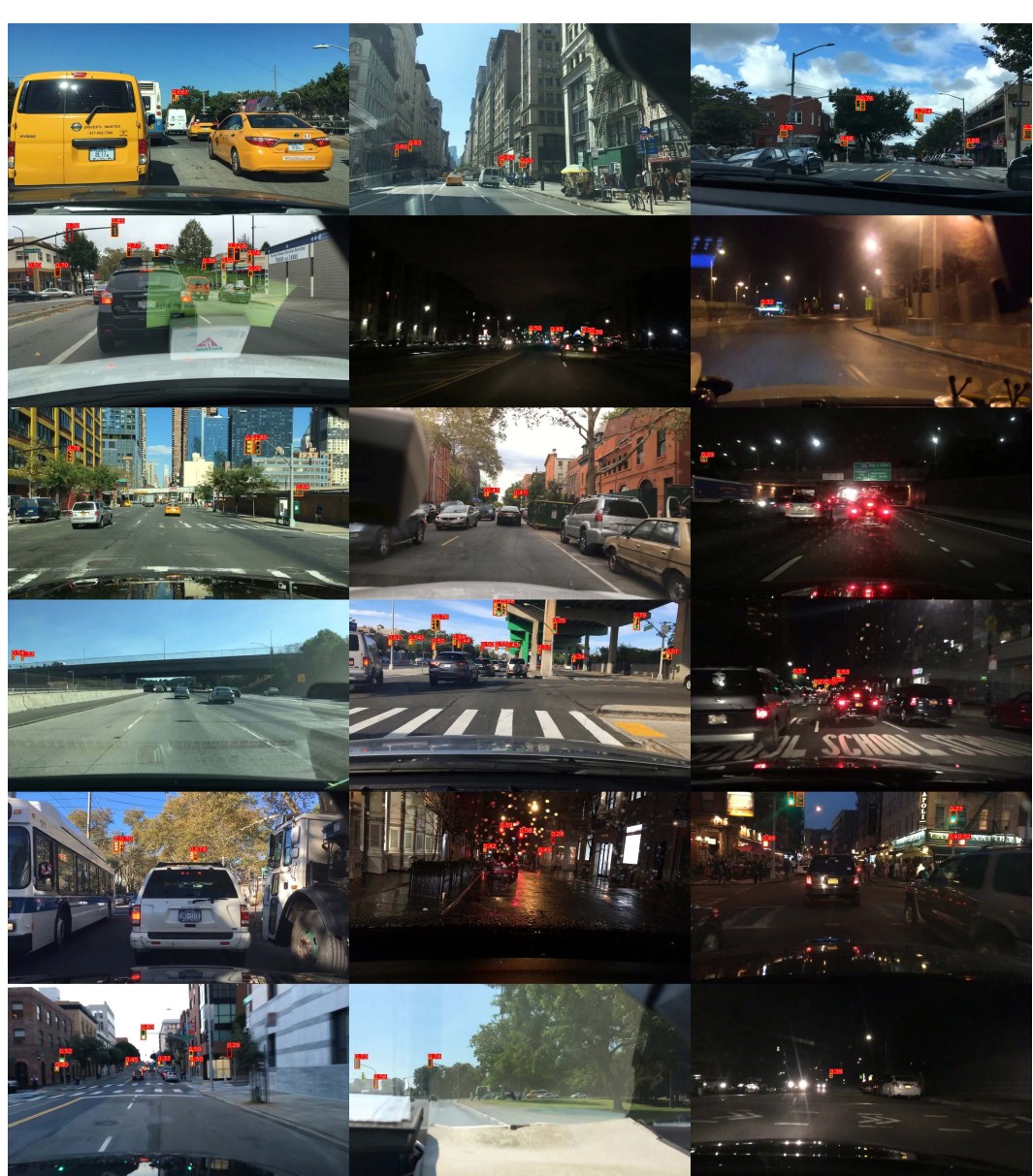

Figure 23: More examples of the category "Traffic Light" of BDD100K sampling from the whole population.

## A.5 TIME COMPARISON

Since the optimization process of our method formulates a non-convex quadratic programming problem and we employ Gurobi optimizer to solve it, it is hard to analyze the time complexity. However, we directly provide a running time comparison. Here we report the running time of different methods on CelebA. The two rows of results correspond to the two categories of CelebA, where the time is measured in seconds. The time consumption of solely constructing the kNN graph is also listed in the last column. We can see that although our method MCSD requires longer running time, the time cost is still generally low and acceptable. Furthermore, in terms of scalability to very large datasets, it is worth noting that our method only requires a validation dataset to work. The validation dataset is essentially a subset sampled from the whole dataset, whose size is much smaller than that of the whole dataset. For example, in CelebA, the validation data size is only 19,867, about 1/10 of the whole dataset size of 202,599. This indicates that for a very large dataset, we could sample a small and appropriate proportion of the whole dataset, and it would be possible for our method to be still effective when being applied to the subset. Besides, the construction of the kNN graph is fast and only takes up a small proportion of running time, which is not the bottleneck of time consumption.

Table 8: Time comparison measured in seconds.

| Blond Hair? | Data Size | Spotlight | Domino | PlaneSpot | MCSD | knn graph |
|:---:|:---:|:---:|:---:|:---:|:---:|:---:|
| Yes | 3,056 | 16.8 | 1.3 | 7.1 | 39.1 | 2.0 |
| No | 16,811 | 93.0 | 26.3 | 45.3 | 171.0 | 13.5 |

## A.6 HYPERPARAMETER SELECTION AND ANALYSES

### A.6.1 HYPERPARAMETER SELECTION

For the hyperparameter of coherence coefficient $\lambda$, we fix it as 1 for experiments on dcbench. For the case studies, we set the search space of $\lambda$ as $\{0.5, 0.8, 1.0, 1.5, 2.0, 2.5, 3.0\}$. We split the validation dataset into two halves, apply our algorithms on one half to obtain a slicing function, apply the slicing function on the other half, and calculate the average performance and manifold compactness of the discovered slice. We choose $\lambda$ that maximizes manifold compactness under the condition that the slice performance is significantly lower than the overall performance, where the threshold can be customized for different tasks. In our experiments, we set it as 15 percent point for accuracy in terms of image classification, and 10 percent point for average precision (AP) in terms of object detection. For the hyperparameter of slice size $\alpha$, in our experiments we set it as 0.05 when the size of the validation dataset is smaller than $5,000$, and set it as 0.01 otherwise. For building kNN graphs, we fix $k = 10$ in our experiments.

### A.6.2 HYPERPARAMETER ANALYSES

In this part, we conduct hyperparameter analyses on the category "Blond Hair" of CelebA for the coherence coefficient $\lambda$, the size $\alpha$, and the number of neighbors $k$ when building the kNN graph. From Table 9, we find that both the accuracy and manifold compactness are best when $\lambda$ and $\alpha$ are in a moderate range, neither too large nor too small. This implies the importance of the balance between pursuing high error and high coherence, which could be achieved by the tuning strategy mentioned in Appendix A.6.1. This also implies the importance of appropriately controlling $\alpha$, i.e. the size of the slice, which is set according to experience in our implementation. Its selection is left for future work.

For $k$, we initially find that $k = 10$ works well and thus fix it. In Table 9 where the manifold compactness of other values has been rescaled to the case of $k = 10$, we can see that the accuracy of the identified slice is generally low compared with the overall accuracy of the blond hair category (76.4%), and the compactness is high when $10 \le k \le 30$. Although $k = 15$ is slightly better than $k = 10$ in terms of compactness, it is still appropriate to select $k = 10$ since it is computationally more efficient.

Table 9: Hyperparameter analyses on the category "Blond Hair" of CelebA for the coherence coefficient $\lambda$, the size $\alpha$, and the number of neighbors $k$. "↑" indicates that higher is better, while "↓" indicates that lower is better. We mark the best method in bold type and underline the second-best. "%" indicates that the digits are percentage values.

| $\lambda$ | Acc. (%)↓ | Comp.↑ | $\alpha$ | Acc. (%)↓ | Comp.↑ | $k$ | Acc. (%)↓ | Comp.↑ |
|---|---|---|---|---|---|---|---|---|
| 0 | 27.8 | 2.94 | 0.005 | 46.2 | 1.00 | 3 | 21.1 | 4.16 |
| 0.5 | 19.6 | 3.36 | 0.01 | **19.2** | 3.31 | 5 | **20.3** | 5.01 |
| 0.8 | **18.1** | 3.71 | 0.03 | 22.8 | 5.84 | 10 | 33.8 | 8.09 |
| 1.0 | 19.6 | 4.85 | 0.05 | 33.8 | **8.09** | 15 | 42.1 | **8.13** |
| 1.5 | 30.1 | 7.14 | 0.1 | 48.9 | 8.07 | 20 | 41.4 | 7.60 |
| 2.0 | 33.8 | **8.09** | 0.15 | 49.4 | 7.60 | 30 | 36.8 | 7.98 |
| 2.5 | 39.9 | 7.93 | 0.2 | 56.6 | 7.40 | 50 | 36.8 | 6.73 |
| 3.0 | 45.1 | 7.99 | 0.3 | 67.7 | 7.54 | 100 | 34.2 | 5.66 |

## A.7 PERFORMANCE IMPROVEMENT VIA UTILIZATION OF THE DISCOVERED ERROR SLICES

We conduct experiments to show performance improvement that the identified error slices could bring via data collection guided by the interpretable characteristics of identified error slices, following the practice of non-algorithmic interventions of Liu et al. (2023). For example, for a given trained image classification model on CelebA, the identified error slice exhibits characteristics of blond hair male, then we could be guided to collect specific data of the targeted characteristics of blond hair male and add to the training data, which is a non-algorithmic intervention and a straightforward and practical way of improving performance of the original model after interpreting characteristics of the identified error slice. Here we compare the results of guided data collection and random data collection. To simulate the guided data collection process, for CelebA, since the identified error slice for the category of blond hair is male and there are extra annotations of sex, we add the images annotated as blond hair male in validation data to training data. Since the identified error slice for not blond hair category is female bearing vintage styles, and there are no related attribute annotations, we directly add the images of the identified slice to the training data. For CheXpert, the identified error slices are from the frontal view for ill patients and from the left lateral view for healthy patients, and CheXpert has annotations of views, so we add the corresponding images in validation data to training data. To simulate the random data collection process, we randomly sample the same number of images from validation data and add to training data for each dataset. Then we retrain the model three times with varying random seeds.

Table 10: Performance of different data collection strategies. "↑" indicates that higher is better. We mark the best strategy in bold type. "%" indicates that the digits are percentage values.

| CelebA | Average Acc. (%)↑ | Worst Group Acc. (%)↑ | CheXpert | Average Acc. (%)↑ | Worst Group Acc. (%)↑ |
|---|---|---|---|---|---|
| Original | 95.3 | 37.8 | Original | 86.7 | 40.3 |
| Random | 95.3±0.4 | 42.4±2.2 | Random | 88.1±0.2 | 50.0±1.1 |
| Guided | **95.4±0.5** | **59.8±3.0** | Guided | **88.7±0.8** | **70.1±1.7** |

Here worst group accuracy is defined following a distribution shift benchmark (Yang et al., 2023), where CelebA and CheXpert are divided into groups according to annotated attributes, and worst group accuracy is an important metric. From Table 10, we can see that guided data collection outperforms the original model and random data collection in both metrics, especially in worst group accuracy. This illustrates that our method is beneficial to performance improvement in practical applications.

## A.8 BENCHMARK DETAILS

Dcbench (Eyuboglu et al., 2022) offers a large number of settings for the task of error slice discovery. Each setting consists of a trained ResNet-18 (He et al., 2016), a validation dataset and a test dataset, both with labels of predefined underperforming slices. The validation dataset and its error slice labels are taken as the input of slice discovery methods, while the test dataset and its error slice labels are used for evaluation.

There are 886 settings publicly available in the official repository of dcbench[1], comprising three types of slices: correlation slices, rare slices, and noisy label slices. The correlation slices are generated from CelebA (Liu et al., 2015), a facial dataset with abundant binary facial attributes like whether the person wears lipstick. Correlation slices include 520 settings. They bear resemblance to subpopulation shift (Yang et al., 2023), where a subgroup is predefined as the minor group by generating spurious correlations between two attributes when sampling training data. That subgroup also tends to be the underperforming group after training. The other two types of slices are generated from ImageNet (Deng et al., 2009), which has a hierarchical class structure. Rare slices include 118 settings constructed by controlling the proportion of a predefined subclass to be small. Noisy label slices include 248 settings formulated by adding label noise to a predefined subclass. Although many settings comprise more than one predefined error slice, we check and find that the given model actually achieves even better performance on a number of slices than the corresponding overall performance. For accurate and convenient evaluation, we select the worst-performing slice of the given model for each setting.

## A.9 EXAMPLES FROM CIVILCOMMENTS

***Warning: Many of these comments are severely offensive or sensitive***

Here in Table 11 we list two parts of comments that are respectively sampled from the slice identified by applying MCSD to the "toxic" category and from all comments of "toxic" category. Since some comments are too long, we do not list the complete comments but additionally list the id of these comments in the dataset for convenience of checking. We check the complete comments and confirm that comments belonging to the error slice identified by MCSD mostly exhibit a positive attitude towards minority groups in terms of gender, race, or religion. This implies that the model tends to treat comments with positive attitudes towards minority groups as non-toxic, while some of these comments are also offensive and toxic.

Table 11: Comments and their id that are respectively sampled from the slice identified by applying MCSD to the "toxic" category and from all comments of "toxic" category. (***Warning: Many of these comments are severely offensive or sensitive***)

| Slice | Content | Id |
|---|---|---|
| MCSD | The Kingdom of Hawai'i has a long, proud history of being the most diverse and inclusive nation... | 5054686 |
| | You have it a bit twisted, TomZ. You say " ...there is evidence that Judge supported same-sex acts ... | 5977193 |
| | scuppers: Go outside. Seriously. You are so invested in this narrative that you're completely losing... | 5865832 |
| | It's not racist to shun people who believe apostates and blasphemers against Islam should be killed... | 6155392 |
| | So, libs have as a leader a person with , IQ less than 70, who can not make a full statement without... | 5316528 |
| | Wow! The US Catholic Church is learning only this year - in 2017 - that racism is rife in the country... | 6320767 |
| | Who is asking for special accommodations here? Transgender people who just want to exist and live... | 5665161 |
| | Poor analogy. Both the KKK and the Blacks are Christians. So cross burning is racial not religious.... | 348794 |
| | Brian Griffin quoted Mencken: "The common man's a fool". Peter Griffin is proof. | 691891 |
| | Wow.... Trump isn't a very deep thinker, and neither is anyone who supports this stupid rule. First... | 5438617 |
| Population | Asian countries for Asians. Black countries for Blacks. but White countries for everybody? That's genocide. | 6299730 |
| | The rapist was a Stanford student; his victim was not. The judge was a Stanford alumnus. We're looking... | 343947 |
| | I wonder if Trump would be in favour of hot black women kneeling? | 6020870 |
| | Plato condemned homosexual relationships as contrary to nature. What are you smoking and where can ... | 5614294 |
| | Black Pride = being black and proud Gay Pride = being gay and proud White Pride = NAZI! | 5815448 |
| | That was Brennan, under Obama, not our good American, Christian president. You really should not make... | 6250038 |
| | So when it's a pretty white woman murdered by her boyfriend, the ANCWL pickets outside the courtroom... | 5763897 |
| | pnw mike, you are right! hillary is a liar. One metric comes from independent fact-checking website... | 520428 |
| | Reminds me of an old Don Rickles joke. "Why do jewish men die before their wives?.....Because they ... | 5215731 |
| | When do see the piece on the worlds most annoying Catholics? Buddhist? Muslims? | 375375 |

---

[1] https://github.com/data-centric-ai/dcbench

## A.10 THEORETICAL ANALYSES

In this subsection, we conduct theoretical analyses on our optimization objective, i.e. Equation (2). First we theoretically prove that the objective not only explicitly considers the manifold compactness inside the identified slice, but also implicitly considers the separability between samples in and out of the identified slice. Then we prove that the optimization objective, where optimized variables are continuous, is equivalent to the discrete version of sample selection with an appropriate assumption. This confirms the validity of our transformation of the problem from the discrete version into the continuous version for the convenience of optimization.

First, we prove a lemma for convenience of later theoretical analyses.

**Lemma 1** *For the inequality constraint $\sum_{i=1}^{n} w_i \leq \alpha n$ in Equation (2), the equality can be achieved for the solution of Equation (2).*

**Proof.** Denote the solution of Equation (2) as $w_1^*, w_2^*, .., w_n^*$. Assume $\sum_{i=1}^{n} w_i^* < \alpha n$. Since $\sum_{i=1}^{n} w_i^* < \alpha n \leq n$, there exists at least one sample weight satisfying $w_k^* < 1$. Let $w_k' = \min\{w_k^* + \alpha n - \sum_{i=1}^{n} w_i^*, 1\}$. We can see that all constraints in Equation (2) are still be satisfied. However, since $w_k' > w_k^*$, the objective has become larger than before. Thus $w_1^*, w_2^*, .., w_n^*$ are not a solution for Equation (2). Since the initial assumption leads to a contradiction, we prove that for the solution of Equation (2), we have $\sum_{i=1}^{n} w_i^* = \alpha n$.

Next, we prove that Equation (2) also implicitly takes the separability between samples in and out of the identified slice by proving that it is equivalent to an objective that explicitly takes the separability into account.

**Proposition 1** *Maximizing $\sum_{i=1}^{n} w_i l_i + \lambda \sum_{i=1}^{n} \sum_{j=1}^{n} w_i w_j q_{ij}$ under the constraints in Equation (2) is equivalent to maximizing $\sum_{i=1}^{n} w_i l_i + \lambda_1 \sum_{i=1}^{n} \sum_{j=1}^{n} w_i w_j q_{ij} - \lambda_2 \sum_{i=1}^{n} \sum_{j=1}^{n} w_i (1 - w_j) q_{ij}$ under the same constraints, where $\lambda = \lambda_1 + \lambda_2$*

**Proof.** Note that since $\{q_{ij}\}_{1 \leq i,j \leq n}$ corresponds to a kNN graph, we have:

$$\sum_{j=1}^{n} q_{ij} = k, \forall 1 \leq i \leq n \tag{3}$$

Combined with Lemma 1, we have:

$$
\begin{aligned}
&\sum_{i=1}^{n} w_i l_i + \lambda_1 \sum_{i=1}^{n} \sum_{j=1}^{n} w_i w_j q_{ij} - \lambda_2 \sum_{i=1}^{n} \sum_{j=1}^{n} w_i (1 - w_j) q_{ij} \\
&= \sum_{i=1}^{n} w_i l_i + (\lambda_1 + \lambda_2) \sum_{i=1}^{n} \sum_{j=1}^{n} w_i w_j q_{ij} - \lambda_2 \sum_{i=1}^{n} \sum_{j=1}^{n} w_i q_{ij} \\
&= \sum_{i=1}^{n} w_i l_i + (\lambda_1 + \lambda_2) \sum_{i=1}^{n} \sum_{j=1}^{n} w_i w_j q_{ij} - \lambda_2 \sum_{i=1}^{n} w_i k \\
&= \sum_{i=1}^{n} w_i l_i + \lambda \sum_{i=1}^{n} \sum_{j=1}^{n} w_i w_j q_{ij} - \lambda_2 \alpha n k
\end{aligned}
\tag{4}
$$

Since $\lambda_2 \alpha n k$ is constant, we have proved the equivalence. Note that the other objective has an extra term of $\sum_{i=1}^{n} \sum_{j=1}^{n} w_i (1 - w_j) q_{ij}$, which exactly represents the separability between samples in and out of the identified slice.

Finally, we prove that Equation (2) is equivalent to the original discrete version of sample selection with proper assumptions.

**Proposition 2** *Assume there exists an ordering of sample index $\{r_i\}_{1 \leq i \leq n}$ satisfying that $q_{r_{i-1}, r_i} = q_{r_i, r_{i-1}} = 0$ and $\alpha \cdot n \in \mathbb{N}^+$. Then there exists a solution of Equation (2) $\mathbf{w}^* = \{w_1^*, w_2^*, .., w_n^*\}$ such that $w_i^* \in \{0, 1\}, \forall 1 \leq i \leq n$.*

**Proof.** Define $J(\mathbf{w}) = \sum_{i=1}^{n} w_i l_i + \lambda \sum_{i=1}^{n} \sum_{j=1}^{n} w_i w_j q_{ij}$. We denotes an optimal solution $\mathbf{w}'$ achieving the optimal value of $J(\mathbf{w})$. Then we can find an optimal solution $\mathbf{w}^*$ satisfying $w_i^* \in \{0, 1\}, \forall 1 \le i \le n$ and $J(\mathbf{w}^*) = J(\mathbf{w}')$.

We initialize $\mathbf{w}^{(0)} = \mathbf{w}'$. Then we sweep a variable $i$ from 1 to $n-1$. For each iteration with $1 \le j \le n-1$, we generate a new weight vector $\mathbf{w}^{(j)}$ by the following process. We can prove that $\mathbf{w}^{(j)}$ is one solution of Equation (2) and $w_{r_i}^{(j)} \in \{0,1\}, \forall 1 \le i \le j$ by mathematical induction, which is already satisfied for $j = 0$.

Firstly, we assign $w_{r_i}^{(j)} = w_{r_i}^{(j-1)}$ for $1 \le i \le j-1$ and $j+2 \le i \le n$ and denote $C = w_{r_j}^{(j-1)} + w_{r_{j+1}}^{(j-1)} \in [0, 2]$.

If $w_{r_j}^{(j-1)} \in \{0,1\}$, we assign $w_{r_j}^{(j)} = w_{r_j}^{(j-1)}$ and $w_{r_{j+1}}^{(j)} = w_{r_{j+1}}^{(j-1)}$.

Otherwise, we reformulate the function $J(\mathbf{w}^{(j)})$ as following (for the sake of brevity, we omit the superscript of $(j)$):

$$
J(\mathbf{w}^{(j)}) = w_{r_j} l_{r_j} + w_{r_{j+1}} l_{r_{j+1}} + \sum_{i \notin \{r_j, r_{j+1}\}} w_i l_i + \lambda \sum_{i \notin \{r_j, r_{j+1}\}} \sum_{s \notin \{r_j, r_{j+1}\}} w_i w_s q_{is}
$$

$$
+ \lambda \left( w_{r_j} \sum_{i \ne r_j} (q_{i,r_j} + q_{r_j,i}) + w_{r_{j+1}} \sum_{i \ne r_{j+1}} (q_{i,r_{j+1}} + q_{r_{j+1},i}) + w_{r_j} w_{r_{j+1}} (q_{r_j, r_{j+1}} + q_{r_{j+1}, r_j}) \right)
$$

$$
= w_{r_j} l_{r_j} + (C - w_{r_j}) l_{r_{j+1}} + \sum_{i \notin \{r_j, r_{j+1}\}} w_i l_i + \lambda \sum_{i \notin \{r_j, r_{j+1}\}} \sum_{s \notin \{r_j, r_{j+1}\}} w_i w_s q_{is}
$$

$$
+ \lambda \left( w_{r_j} \sum_{i \ne r_j} (q_{i,r_j} + q_{r_j,i}) + (C - w_{r_j}) \sum_{i \ne r_{j+1}} (q_{i,r_{j+1}} + q_{r_{j+1},i}) + w_{r_j} (C - w_{r_j})(q_{r_j, r_{j+1}} + q_{r_{j+1}, r_j}) \right)
\tag{5}
$$

We can see that $J(\mathbf{w}^{(j)})$ is a quadratic or linear function with respect to $w_{r_j}^{(j)}$. Since $q_{r_j, r_{j+1}} = q_{r_{j+1}, r_j} = 0$, $J(\mathbf{w})$ becomes a linear function of $w_{r_j}^{(j)}$.

Because setting $w_{r_j}^{(j)} = w_{r_j}^{(j-1)} \in (0, 1)$ is a global minimum, the coefficient of $J(\mathbf{w})$ with respect to $w_{r_j}^{(j)}$ equals zero. Thus $J(\mathbf{w})$ is constant with respect to $w_{r_j}^{(j)}$. Therefore, we set the value of $w_{r_j}^{(j)}$ and $w_{r_{j+1}}^{(j)}$ as the following two rules:

- If $1 \le C < 2$, we assign $w_{r_j}^{(j)} = 1$ and $w_{r_{j+1}}^{(j)} = C - 1$
- If $0 < C < 1$, we assign $w_{r_j}^{(j)} = 0$ and $w_{r_{j+1}}^{(j)} = C$

It is obvious that $J(\mathbf{w}^{(j)}) = J(\mathbf{w}^{(j-1)})$. Therefore, $\mathbf{w}^{(j)}$ also achieves the optimal value for Equation (2). Since $w_{r_i}^{(j)} = w_{r_i}^{(j-1)} \in \{0,1\}, \forall 1 \le i < j$ and $w_{r_j}^{(j)} \in \{0,1\}$, we conclude that $w_{r_i}^{(j)} = w_{r_i}^{(j-1)} \in \{0,1\}, \forall 1 \le i \le j$.

Finally, we obtain $\mathbf{w}^{(n-1)}$ where we have $w_{r_i}^{(n-1)} \in \{0,1\}, \forall 1 \le i \le n-1$. Since the sum of $\mathbf{w}^{(n-1)}$ is an integer $\alpha n$, $w_{r_n}^{(n-1)}$ in also an integer. According the construction of $w_{r_n}^{(n-1)}$ in the above two rules, it can be found that $0 \le w_{r_n}^{(n-1)} \le 1$. Therefore, we have $w_i^{(n-1)} \in \{0,1\}, \forall 1 \le i \le n$.

The pursued solution of $\mathbf{w}^*$ can be obtained by setting $\mathbf{w}^* = \mathbf{w}^{(n-1)}$.

**Remark** Since $n >> k$, it is likely that the constructed graph is extremely sparse. Therefore, it is easy to find a sample ordering that the contiguous samples are not connected, which means that our assumption is satisfied. This proposition proves the equivalence between our continuous optimization formulation to the discrete sample selection.

## A.11 ABLATION OF THE SLICING FUNCTION

In Algorithm 1, after acquiring the desired samples, we additionally train an MLP as the slicing function. This is because the standard evaluation process of error slice discovery, an error slice discovery method is applied to validation data to obtain the slicing function, and then the slicing function is applied to test data to calculate evaluation metrics and conduct case analyses. Such practice is also adopted by dcbench (Eyuboglu et al., 2022), the benchmark of error slice discovery. Thus we follow this practice by additionally train a slicing function after we obtain the desired samples, so that we can compare fairly with previous methods. However, even without training this slicing function, our method can still produce meaningful results of case studies. We show examples randomly sampled from optimized results of Equation (2) in Figure 24 and 25. We can see that there is still high coherence in the identified slices.

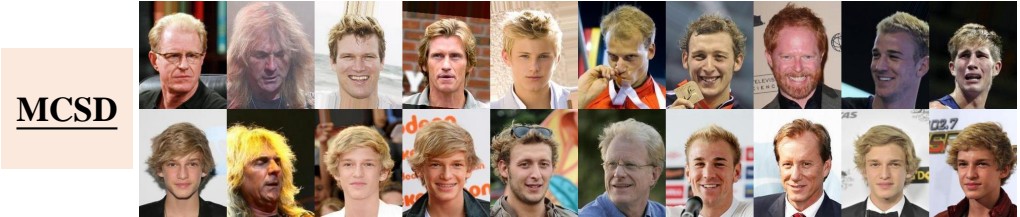

Figure 24: Examples of the category "blond hair" of CelebA directly sampling from optimized results of Equation (2).

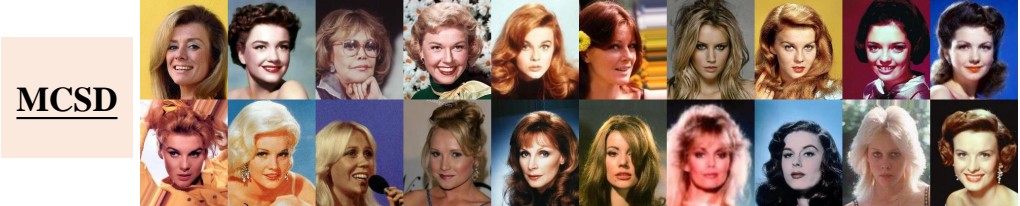

Figure 25: Examples of the category "not blond hair" of CelebA directly sampling from optimized results of Equation (2).

## B  Related Work

**Subpopulation Shift**   It is widely acknowledged that models tend to make systematic mistakes on some subpopulations, which leads to the problem of subpopulation shift (Yang et al., 2023). To guarantee the worst subpopulation performance, some generate pseudo environment labels (Creager et al., 2021; Nam et al., 2021) and then apply existing invariant learning methods (Arjovsky et al., 2019; Krueger et al., 2021). Others take advantage of importance weighting to upweight the minority group or worst group like GroupDRO (Sagawa et al., 2019) and JTT (Liu et al., 2021), or to combine with Mixup (Zhang et al., 2018) for more benefits (Han et al., 2022). A more recent work points out that current methods for subpopulation shifts heavily rely on the availability of group labels during model selection, and even simple data balancing techniques can achieve competitive performance (Idrissi et al., 2022). For better comparisons between algorithms and promotion of future algorithm development, Yang et al. (Yang et al., 2023) establish a comprehensive benchmark across various types of datasets.

**Error Slice Discovery**   Instead of developing algorithms to improve subpopulation robustness, the operation of error slice discovery has also attracted much attention recently. It is more flexible in that it can be followed by either non-algorithmic interventions like collecting more data for error slices, or algorithmic interventions like upweighting data belonging to error slices. There are mainly two paradigms for the process of error slice discovery. The first paradigm, also the more traditional practice, separates error slice discovery and later interpretation via case analyses or with the help of multi-modal models. Spotlight (d'Eon et al., 2022) attempts to learn a centroid in the representation space and employ the distance to this centroid as the error degree. InfEmbed (Wang et al., 2023b) employs the influence of training samples on each test sample as embeddings used for clustering. PlaneSpot (Plumb et al., 2023) concatenates model prediction probability with dimension-reduced representation for clustering. All of them interpret the identified slices via case analyses directly. Meanwhile, the error-aware Gaussian mixture algorithm Domino (Eyuboglu et al., 2022) is followed by finding the best match between candidate text descriptions and the discovered slice in the representation space of multi-modal models like CLIP (Radford et al., 2021). This paradigm has a relatively high requirement for the coherence of identified error slices so that they can be interpreted. The second paradigm incorporates the discovery and interpretation of error slices together. Both HiBug (Chen et al., 2023) and PRIME (Rezaei et al., 2024) divide the whole population of data into subgroups through proposing appropriate attributes and conducting zero-shot classification for these attributes using pretrained multi-modal models, and then directly calculate average performance for subgroups to identify the risky ones. The obtained subgroups are naturally interpretable via the combination of the attribute pseudo labels. Two recent works (Wiles et al., 2022; Gao et al., 2023) generate data from diffusion models (Rombach et al., 2022) before identifying error slices, avoiding the requirement of an extra validation dataset for slice discovery. It is obvious that the second paradigm heavily relies on the quality of proposed attributes and the capability of pretrained multi-modal models.

**Error Prediction**   Another branch of works sharing a similar goal with error slice discovery is error prediction (or performance prediction). Although they are also able to find slices with high error, they focus on predicting the overall error rate given an unlabeled test dataset, and measure the effectiveness of error prediction methods via the gap between the predicted performance and the ground-truth one. Moreover, they do not emphasize the coherence and interpretability of error slices. Currently, there are several ways for error prediction. Some employ model output properties on the given test data like model confidence (Garg et al., 2021; Guillory et al., 2021), neighborhood smoothness (Ng et al., 2022), prediction dispersion (Deng et al., 2023), invariance under transformations (Deng et al., 2021), etc. Inspired by domain adaptation (Long et al., 2015; Ben-David et al., 2010), some make use of distribution discrepancy between training data and unlabeled test data (Deng & Zheng, 2021; Yu et al., 2022; Lu et al., 2023). Others utilize model disagreement between two models identically trained except random initialization and batch order during training (Jiang et al., 2021; Baek et al., 2022; Chen et al., 2021; Kirsch & Gal, 2022), which exhibits SOTA performance in the error prediction task (Trivedi et al., 2023).

