# OpenReview forum: "Error Slice Discovery via Manifold Compactness"
_ICLR.cc/2025/Conference — Submitted to ICLR 2025_

### Official Review · Reviewer_Dag2 · 2024-10-28

**Soundness:** 2
**Presentation:** 3
**Contribution:** 3
**Rating:** 5
**Confidence:** 4

**Summary:**

The paper identifies error slices within data subsets using manifold compactness and measures the similarity of certain features in the error samples based on the level of manifold compactness. This approach focuses on detecting error slices that have both high risk (high error rate) and high coherence, making these slices easier to interpret and understand.

**Strengths:**

1. It provides a different perspective for measuring error slices, and this metric does not require predefined slice label information.
2. An algorithm called MCSD is proposed, which directly uses model risk and coherence as optimization objectives.
3. The experimental data is thorough and demonstrates the effectiveness of the proposed method across multiple datasets.

**Weaknesses:**

The further explanation of the experiments is missing. More information pls see Questions.

**Questions:**

1. In the first paragraph of the problem definition on page 2: Does $\mathcal{Y} \times \mathcal{Y}$ represent the Cartesian product of two sets? If so, one should represent the predicted label, and the other should represent the true label. Why are both symbols $\mathcal{Y}$?

2. The method proposed by the authors provides greater interpretability and feature similarity consistency when identifying error subsets, as shown in the visualization results of the CELEBA dataset in Figure 4. However, why does the ACC of the error slices identified by MCSD outperform the SOTA methods by such a large margin? Besides this specific similar feature, what other factors could lead to poor performance of the sliced subsets?

3. Why does Table 4 only compare the MCSD method with the Spotlight method?

4. I have some concerns about the metric proposed by the authors. The expected trend in the experimental data is that as ACC increases, the Comp result should also increase. Why do Tables 2, 3, and 5 not follow this trend? Perhaps I am missing something—do the authors provide any further explanation?

---

> ### Author Response · Authors · 2024-11-22
>
> Thanks for your questions! We try to address your concerns below.
>
> **Q1**: Does $\mathcal{Y}\times\mathcal{Y}$  represent the Cartesian product of two sets? If so, one should represent the predicted label, and the other should represent the true label. Why are both symbols $\mathcal{Y}$?
>
> **A1**: Thanks for your question. Yes, it represents the Cartesian product of two sets, and they respectively represent the predicted label and the true label. In Line 73, we denote the prediction model as $f_{\theta}: \mathcal{X}\mapsto\mathcal{Y}$, so the symbol $\mathcal{Y}$ actually represents the predicted label space instead of the true label space. This does not contradict the notation in Line 71 $Y\in\mathcal{Y}$ since the true label space is a subset of the predicted label space. For example, for classification tasks of $K$ categories, the predicted label space is $\Delta^{K-1}=\\{(p_1, p_2,...,p_K)|\sum_{i=1}^k p_i=1, \forall p_i\geq 0, i=1,2,...,K\\}$, while the true label space is all $K$ dimensional one-hot vectors. For regression tasks, both the predicted label space and the true label space are $\mathbb{R}$. Meanwhile, such a notation is also employed in previous works that are influential in their own areas [1,2,3], including error slice discovery, domain generalization, and continual learning.
>
> **Q2-1**: Why does the ACC of the error slices identified by MCSD outperform the SOTA methods by such a large margin?
>
> **A2-1**: Thanks for your question. As we mentioned in Definition 1 and Line 312-318, although there are two evaluation metrics at the same time, since we put more emphasis on the coherence inside the identified slice for better interpretability, we only require the performance of the identified slice to be lower than the overall average performance by a certain threshold, while expecting the coherence metric to be as high as possible. Thus the higher accuracy of the error slices identified by MCSD than previous SOTA methods is not our concern, since the accuracies of error slices identified by both MCSD and previous SOTA methods are significantly lower than the overall performance.
>
> **Q2-2**: Besides this specific similar feature, what other factors could lead to poor performance of the sliced subsets?
>
> **A2-2**: Thanks for your question. We would like to clarify that we are not trying to find similar features thus to decrease the performance of the identified slices, but we are trying to find slices with similar features that are underperforming at the same time. Thus it might be hard to say there is a causal relationship between poor performance of slices and similar features.
>
> **Q3**: Why does Table 4 only compare the MCSD method with the Spotlight method?
>
> **A3**: Thanks for your question. Table 4 presents the results on a task of object detection. As stated in Line 490-491, we could not compare with Domino or PlaneSpot since neither of them is applicable to tasks other than classification. This is because they require the prediction probability as the input, thus only applicable to classification, while our algorithm takes the prediction loss as input, naturally more flexible and applicable to various tasks. Such flexibility is an advantage of our method, which is empirically demonstrated through experiments of various case studies.
>
> **Q4**: The expected trend in the experimental data is that as ACC increases, the Comp result should also increase. Why do Tables 2, 3, and 5 not follow this trend?
>
> **A4**: Thanks for your question. We would like to clarify that our goal is to find underperforming slices that are coherent and interpretable, and both the risk (reversely corresponding to "ACC") and coherence (corresponding to "Comp") of the slice are part of our optimization objective (Equation 2). They are simultaneously optimized, and there is no expected trend for one of them when the other increases or decreases.
>
>
>
> ### References
>
> [1] Eyuboglu et al. "Domino: Discovering Systematic Errors with Cross-Modal Embeddings." ICLR, 2022.
>
> [2] Zhou et al. "Domain generalization: A survey." TPAMI, 2022.
>
> [3] Lopez-Paz et al. "Gradient episodic memory for continual learning." NeurIPS, 2017.

---

> ### Author Response · Authors · 2024-12-01
>
> Dear Reviewer Dag2,
>
> We would like to thank you for your time and efforts in reviewing our paper! We really look forward to your response and we would be very glad to engage in any further discussion if there are remaining concerns in the last three days of the discussion period.
>
> Meanwhile, we also sincerely hope that you might take the revisions mentioned in our common responses to all reviewers into account when evaluating our contributions, including the theoretical analyses of our optimization objective, the newly added comparisons with more metrics directly calculated in Euclidean space, and other modifications for better clarity.
>
> Thanks again for your efforts during the reviewing process!

---

> ### Author Response · Authors · 2024-12-03
>
> Dear Reviewer Dag2,
>
> We sincerely apologize for sending another message.
>
> We greatly appreciate your time and efforts in reviewing our work, and we look forward to your response very much! Meanwhile, we kindly wish to remind that there are less than 12 hours before Dec 2nd AOE, after which we could not exchange responses with each other, and we are really willing to engage in any further discussion to address your concerns.
>
> Thank you again for taking time to review!

---

> ### Author Response · Authors · 2024-12-03
>
> Dear Reviewer Dag2,
>
> We greatly thank you for your time in the review! We would like to kindly remind that there is only about an hour left for reviewers and authors to exchange responses with each other, and we are really eager to receive your response. We always look forward to discussing with you and will try our best to address any of your concerns.
>
> Thanks again!

---

### Official Review · Reviewer_jVFS · 2024-10-29

**Soundness:** 2
**Presentation:** 3
**Contribution:** 2
**Rating:** 5
**Confidence:** 4

**Summary:**

In this paper, authors propose manifold compactness, a coherence metric that does not depend on additional information, by incorporating data geometric properties into its design. And experiments are conducted on typical datasets to empirically validate the reasonableness of the metric. Authors then propose an method (MCSD), which directly considers risk and coherence as optimization objectives.

**Strengths:**

The authors show the results of many experiments.

**Weaknesses:**

There are a number of problems that the author needs to address, as detailed in the Questions section.

**Questions:**

1. The example given by the author on page 4 is not convincing. The author claims that monotonicity can be used as a criterion for determining whether a measure of the semantics of slices is good, yet I do not think this is sufficient. Suppose I propose a measure that uses the difference between the number of elements in the cluster and the total number of elements as the measure, in which case the slices, as shown in Figure 3, would result in a positive increase in the value of the measure for each further slice, with a consistent trend with the measure proposed by the authors. Clearly, this arbitrarily proposed measure is not superior. Therefore, the authors need to further clarify the example.
(e.g. Provide step-by-step details on how Figure 2c is generated, or clarify the relationship between the optimization variables w and g and how they relate to generating a new space)

2. Some key details are missing. The variables involved in the optimization objective appear to be w and the binary classifier g; how does this generate an new space? How do you generate the picture that the authors show in Figure 2c? If my understanding is wrong, please let me have further information.


3. The author mentions in line 273 that the w obtained by solving Eq. 2 is used to train the slice function g. Are the training samples obtained using w already the desired samples? Does this result in a meaningless slice function g?
(e.g. Clarify the purpose of training g if w already identifies the desired samples)


4. Does the slicing function obtained by the authors give the same output results from the same input data in each slicing? If so, does multiple slices require re-solving Eq. 2 and re-training the slice function g? I think the desired slicing result can already be obtained before training the slicing function g, does this mean that training the slicing function is meaningless? Can the authors clarify this?

5. What is the difference between Eq. 2 and a clustering method? Equation 2 seems to be a clustering method as well.
(e.g. Discuss how the proposed method specifically addresses the goals of error slice discovery in ways that clustering alone may not )

6. In line 71, the authors describe the coherence method of past methods as only indirectly joining the optimization process. However, the authors' method also seems to be a two-step optimization process: i.e., first optimize the clustering and then optimize the slicing function g. If I have misunderstood, please clarify further.

7. The authors need to clarify further in the Section 1 what is the reason for proposing this method. The current motivation of the authors for proposing the method seems to be somewhat weak.

---

> ### Author Response · Authors · 2024-11-22
>
> Thanks for your questions! We try to address your concerns below.
>
> **Q1**: Use of monotonicity as a criterion for determining whether a measure of the semantics of slices is good.
>
> **A1**: Thanks for your feedback. We would like to clarify that the monotonicity is one of the criteria for determining the coherence metric, but not the only one. Two implicit but necessary requirements are: being normalized by the sample size in the slice, and being intuitively reasonable. Both our proposed metric and variance satisfy these two requirements, so we did not emphasize these in our paper, and focused on the comparison in terms of monotonicity. The example that "uses the difference between the number of elements in the cluster and the total number of elements as the measure" might not be appropriate since it is not normalized by the sample size, and it is obviously not a reasonable metric to depict the coherence. Besides, in our revised version of paper, in Appendix A.1.2 on page 16, we also add comparisons with more coherence metrics that are directly calculated in the Euclidean space to further demonstrate the validity of our proposed metric. More details could be referred to in our **first common response** to all reviewers and the revised paper pdf.
>
> **Q2-1**: The variables involved in the optimization objective appear to be w and the binary classifier g. How does this generate a new space?
>
> **A2-1**: Thanks for your question. We would like to clarify that the space of the manifold is the constructed knn graph, and the "distance" between sample $i$ and $j$ is $q_{ij}$, i.e. whether there exists an edge between node $i$ and $j$ in the knn graph. In other words, the space has been pre-calculated and fixed when optimizing w or g.
>
> **Q2-2**: How do you generate the picture that the authors show in Figure 2c?
>
> **A2-2**: Thanks for your question. The details of generating Figure 2c have already been described in Appendix A.1.1. We employ features extracted by the image encoder of CLIP-ViT-B/32 and employ UMAP [1], a well-known dimension-reduction method that is manifold-based, to reduce the 512-dimension features into 2 dimensions and visualize them, with the blue dots represent correct samples and the red dots represent error samples. Besides Figure 2c, the result of Figure 2b also comes from a manifold-based dimension-reduction method t-SNE [2], while the result of Figure 2a comes from PCA that mainly preserves pairwise Euclidean distances between data points. We can see that the visualization of t-SNE and UMAP shows much clearer clustering structures than that of PCA, either having a larger number of clusters or exhibiting larger margins between clusters. This indicates that it could be better to measure coherence in the metric space of a manifold than in the original Euclidean space.
>
> **Q3**: Why do we need to train the slicing function g when already obtaining the desired samples?
>
> **A3**: Thanks for your question. In fact, **the slicing function g is not really a necessary component of our algorithm**. It is mainly designed to align with previous settings of evaluation and provide fair comparisons with previous methods. In the standard evaluation process of error slice discovery, an error slice discovery method is applied to validation data to obtain the slicing function, and then the slicing function is applied to test data to calculate evaluation metrics and conduct case analyses. Such practice is also adopted by dcbench [3], the benchmark of error slice discovery. If there is no test data and the evaluation metrics are calculated on validation data, which is already used to fit the slicing function, it is possible to achieve good performance via overfitting the validation data. To follow this practice, we additionally train a slicing function after we obtain the desired samples, so that we can compare fairly with previous methods. Since it is not a necessary component, for a better understanding of our algorithm, in our revised version of paper, we remove the training of the slicing function from our algorithm procedure on page 5, and put it into the section of experiments as a part of experimental details. It is worth noting that even without training this slicing function, our method can still produce meaningful results of case studies. We add the images sampled from the examples obtained before training the slicing function in Appendix A.11 on page 36 in our revised version of paper. We can see that our method is still capable of finding coherent slices.

---

> > ### Author Response · Authors · 2024-11-22
> >
> > **Q4-1**: Does the slicing function obtained by the authors give the same output results from the same input data in each slicing?
> >
> > **A4-1**: Yes, it does.
> >
> > **Q4-2**: If so, does multiple slices require re-solving Eq. 2 and re-training the slice function g?
> >
> > **A4-2**: Yes, they do. The showcase for multiple error slices has been provided in Appendix A.2 both versions of paper.
> >
> > **Q5**: What is the difference between Eq. 2 and a clustering method? Equation 2 seems to be a clustering method as well.
> >
> > **A5**: Thanks for your question. Clustering methods tend to indirectly incorporate the constraint of coherence into their design and aim to find all error slices at a time, which is too ambitious and hard to achieve effectively. Our method directly incorporates manifold compactness, the coherence metric, into the optimization objective, and focuses on one slice at a time (but can be used to find multiple slices as illustrated in Appendix A.2). It might also be seen as a clustering method in a broad sense, but it differs from previous clustering-based error slice discovery methods as we explained above.
> >
> > **Q6**: The authors describe the coherence method of past methods as only indirectly joining the optimization process. However, the authors' method also seems to be a two-step optimization process.
> >
> > **A6**: Thanks for your feedback. The meaning of directly joining the optimization process refers to the direct incorporation of coherence term, i.e. the second term in Equation 2, into the optimization objective, which is the core component of our method. As stated in **A3** (our answer to **Q3**), the training of the slicing function is not a necessary component of our algorithm, and in our revised version of paper we remove the training of the slicing function from our algorithm procedure on page 5, and put it into the section of experiments as a part of experimental details.
> >
> > **Q7**: The authors need to clarify further in the Section 1 what is the reason for proposing this method. The current motivation of the authors for proposing the method seems to be somewhat weak.
> >
> > **A7**: Thanks for your feedback. In our revised version of paper, we modify Section 1 to better illustrate the motivation of error slice discovery in the first paragraph. As for the motivation of our method, we would like to clarify that **the motivation of our proposed method is closely related to proposing the metric of manifold compactness**. Previously, although coherence is pivotal to error slice discovery, there is no proper metric of slice coherence without relying on extra information like predefined metadata. The evaluation process strongly relies on the quality of metadata, and previous metrics cannot be directly incorporated into the optimization objective. Thus **the motivation of our metric and method is: (1) Getting rid of the requirement of predefined metadata when measuring the coherence and evaluate error slice discovery algorithms; (2) Directly incorporating the coherence metric into the optimization objective to improve the effectiveness of error slice discovery**. Motivated by these two points, and inspired by the data geometry property that high dimensional data tends to lie on a low-dimensional manifold [4,5,6], we propose the coherence metric of manifold compactness without relying on predefined metadata and design our method by incorporating this metric into the optimization objective to formulate a quadratic programming problem.
> >
> >
> >
> > ### References
> >
> > [1] McInnes et al. "Umap: Uniform manifold approximation and projection for dimension reduction." arXiv, 2018.
> >
> > [2] Van der Maaten et al. "Visualizing data using t-SNE." JMLR, 2008.
> >
> > [3] Eyuboglu et al. "Domino: Discovering Systematic Errors with Cross-Modal Embeddings." ICLR, 2022.
> >
> > [4] Belkin et al. "Laplacian eigenmaps for dimensionality reduction and data representation." Neural computation, 2003.
> >
> > [5] Roweis et al. "Nonlinear dimensionality reduction by locally linear embedding." Science, 2000.
> >
> > [6] Tenenbaum et al. "A global geometric framework for nonlinear dimensionality reduction." Science, 2000.

---

### Official Review · Reviewer_q2Dt · 2024-10-29

**Soundness:** 3
**Presentation:** 3
**Contribution:** 3
**Rating:** 8
**Confidence:** 4

**Summary:**

The paper focuses on error slice discovery, where error slices refer to specific data subsets on which the model underperforms. It first proposes a new evaluation metric, manifold compactness, which does not rely on external information such as predefined slices or metadata attributes. Furthermore, the paper leverages manifold compactness to design a new error slice discovery algorithm, which can be flexibly applied across various task models.

**Strengths:**

1. This is a relatively new and interesting field that merits further exploration.
2. The organization and writing of this paper are reader-friendly.
3. Extensive case studies support the authors' views, accompanied by new metrics and algorithms.

**Weaknesses:**

1. Although extensive case studies support the authors' viewpoints, the lack of theoretical analysis is regrettable.
2. There is no analysis of time and space complexity. For a dataset of size $n$, constructing a graph requires at least $O(n^2) $ and solving Equation 2 takes $O(n^3)$. In fact, as seen in Table 8, efficiency is indeed a drawback. The authors might consider explaining why this efficiency loss is worthwhile.
3. The paper suggests that Euclidean distance is inferior to manifold, but this may be misleading. In fact, constructing the k-nearest neighbors graph still depends on Euclidean distance. Additionally, because distance variance lacks bounds, it is more sensitive to samples, especially if the sample features are not normalized. Therefore, the superiority of manifold compactness over variance may not be fully convincing. Consider adding a new distance-based measure. Different manifold constructions and distance calculations are also anticipated.
4. The algorithm is divided into multiple parts, and an end-to-end model would be more desirable.

**Questions:**

See Weaknesses.

---

> ### Author Response · Authors · 2024-11-22
>
> Thanks for your valuable feedback! We try to address your concerns below.
>
> **Q1**: The lack of theoretical analysis is regrettable.
>
> **A1**: Thanks for pointing this out. Our formulated optimization objective (Equation 2) is a non-convex quadratic programming problem. In our revised version of paper, we conduct some theoretical analyses of it. First, we theoretically prove that our optimization objective not only explicitly considers the manifold compactness inside the identified slice, but also implicitly considers the separability between samples in and out of the identified slice. Second, we theoretically prove that the final optimization objective, where optimized variables are continuous, is equivalent to the original discrete version of sample selection with proper assumptions. This confirms the validity of our transformation of the problem from the discrete version into the continuous version, which is more convenient for optimization. For detailed formulations and proof, please refer to Appendix A.10 on page 34-35 in our revised version of paper.
>
> **Q2**: There is no analysis of time and space complexity. For a dataset of size $n$, constructing a graph requires at least O($n^2$) and solving Equation 2 takes O($n^3$). In fact, as seen in Table 8, efficiency is indeed a drawback. The authors might consider explaining why this efficiency loss is worthwhile.
>
> **A2**: Thanks for your feedback. For constructing the knn graph, the time complexity is $O(dn{\rm log}n)$ instead of $O(n^2)$, where $d$ denotes the feature dimension and $n$ denotes the sample size, since it can be constructed based on k-d tree. We can see in Table 8 that its construction is fast and only takes up a small proportion of running time. For solving Equation 2, as we stated at the beginning of Appendix A.5, since our optimization objective is a non-convex quadratic programming problem, and we employ Gurobi, a powerful and popular mathematical optimization solver designed for such problems, it is hard to analyze the time complexity. Therefore, we directly conduct empirical time comparisons between different methods in Table 8. Considering that error slice discovery serves as extra analyses instead of traditional model training or inference that sometimes poses a high demand on efficiency, and that our time cost is still generally low within only several minutes, we believe such an increase of time cost is acceptable and worthwhile given the effectiveness and flexibility of our method, which we have demonstrated through comprehensive experiments.
>
> For the space complexity, the main increment compared with previous methods is the knn graph. Although the brute-force implementation requires space of $O(n^2)$ , with the help of k-d tree, which avoids storing all pairwise distances, the storage becomes $O(nd)$ (storage of the k-d tree) plus $O(nk)$ (storage of the sparse knn graph). Such a cost of space is also acceptable.
>
> **Q3-1:** The paper suggests that Euclidean distance is inferior to manifold, but this may be misleading. In fact, constructing the k-nearest neighbors graph still depends on Euclidean distance.
>
> **A3-1**: Thanks for your feedback. A basic assumption of manifold learning is that it shares similar local properties with the Euclidean space, i.e. it can be approximated by Euclidean space locally or in small regions [1,2]. Thus it is reasonable to use connections between close points to depict the manifold. Meanwhile, the construction of knn graph is popular and widely adopted in manifold learning [3,4,5].

---

> ### Author Response · Authors · 2024-11-22
>
> **Q3-2**: Additionally, because distance variance lacks bounds, it is more sensitive to samples, especially if the sample features are not normalized. Therefore, the superiority of manifold compactness over variance may not be fully convincing. Consider adding a new distance-based measure. Different manifold constructions and distance calculations are also anticipated.
>
> **A3-2**: Thanks for your valuable suggestion! Indeed variance could be sensitive to outliers, but it is not appropriate to normalize sample features here since they are extracted by pretrained models and the scale difference across feature dimensions could be meaningful, unlike in tabular data where normalization is standard practice. To mitigate the issue of sensitivity to outliers, we add the comparison with Mean Absolute deviation (MeanAD), Median Absolute Deviation (MedianAD), and Interquartile Range (IQR), which are less sensitive to outliers in Appendix A.1.2 on page 16 in our revised version of paper. We can see that these metrics that are directly calculated in the Euclidean space show similar phenomenons to variance, i.e. the metric value of the more coarse-grained slice is sometimes even smaller than that of the more fine-grained slice, which contradicts our expectation. As for manifold constructions, we choose knn graph not only because it is popular and relatively efficient [3,4,5], but also because a graph learning method is convenient when formulating the optimization objective (Equation 2) as a standard optimization problem.
>
> **Q4**: The algorithm is divided into multiple parts, and an end-to-end model would be more desirable.
>
> **A4**: Thanks for your suggestion! Our algorithm was divided into these parts: extracting image features, building a knn graph, solving the non-convex quadratic programming problem, and training a slicing function. For the training of the slicing function, the main reason of requiring it is to follow the standard setting of the error slice discovery benchmark dcbench [6] and provide fair comparisons with previous methods. In our revised version of paper, we remove this part from our algorithm procedure on page 5 and put it into the section of experiments as a part of experimental details. We also add a new subsection of Appendix A.11 on page 36 to demonstrate that our method is still effective without additionally training the slicing function. As for extracting image features, it is common practice employed in previous error slice discovery literature [6,7,8]. As for building a knn graph, we would leave the development of an end-to-end algorithm that incorporates manifold learning and error slice discovery  for future work.
>
>
>
> ### References
>
> [1] Lee et al. "Smooth manifolds". Springer New York, 2012.
>
> [2] Do Carmo, Manfredo P. "Differential geometry of curves and surfaces". Courier Dover Publications, 2016.
>
> [3] Zemel et al. "Proximity graphs for clustering and manifold learning." NeurIPS, 2004.
>
> [4] Pedronette et al. "Unsupervised manifold learning through reciprocal kNN graph and Connected Components for image retrieval tasks." Pattern Recognition, 2018.
>
> [5] Dann et al. "Differential abundance testing on single-cell data using k-nearest neighbor graphs." Nature Biotechnology, 2022.
>
> [6] Eyuboglu et al. "Domino: Discovering Systematic Errors with Cross-Modal Embeddings." ICLR, 2022.
>
> [7] Jain et al. "Distilling Model Failures as Directions in Latent Space." ICLR, 2023.
>
> [8] Plumb et al. "Towards a More Rigorous Science of Blindspot Discovery in Image Classification Models." TMLR, 2023.

---

> > ### Comment · Reviewer_q2Dt · 2024-11-27
> >
> > Thanks for your response and resolving most of my concerns. Regarding theoretical analysis (Q1), I am more eager to understand why manifold compactness works and where its limitations lie. Regarding Q3, some statements in the paper should be more rigorous, and it cannot be directly said that manifolds are superior to Euclidean distance, because the construction of manifolds depends on Euclidean distance, but it can be said that they are superior to the variance of distance. Overall, I am inclined to accept this paper as the empirical research is particularly solid, but I still hesitate about whether to increase my rating. I hope you can further explain the value of the paper, including but not limited to its applications and subsequent research.

---

> > > ### Author Response · Authors · 2024-11-28
> > >
> > > Thanks for your response! We greatly appreciate your recognition, and we are glad to engage in the discussion. Here we try to address your remaining concerns as follows:
> > >
> > > **Q5**: Regarding theoretical analysis (Q1), I am more eager to understand why manifold compactness works and where its limitations lie.
> > >
> > > **A5**: Thanks for your question! In terms of why manifold compactness works, it captures the data geometry property that **high dimensional data tends to lie on a low-dimensional manifold, which is an inherent assumption instead of a deduced theorem, and has been widely accepted by literature of manifold learning**. The theoretical analyses of previous manifold learning literature [1-5] actually focus on the properties of their algorithm and optimization. Thus we devote our efforts to conducting theoretical analyses of our optimization objective while justifying manifold compactness via the empirical studies in Section 3 and Appendix A.1. In terms of its limitations, the metric of manifold compactness might not capture all complex semantic patterns with different granularities. The capability boundary of our metric and our method is left for future work.
> > >
> > > **Q6**: Some statements in the paper should be more rigorous, and it cannot be directly said that manifolds are superior to Euclidean distance, because the construction of manifolds depends on Euclidean distance, but it can be said that they are superior to the variance of distance.
> > >
> > > **A6**: Thanks for your suggestion! **We have made modifications of related expressions marked in color orange in our second revision of the paper**, where we change the compared object from "Euclidean distance base metrics" to "metrics that are directly calculated in Euclidean space", and "measure coherence in Euclidean space" to "measure coherence via metrics directly calculated in Euclidean space". Here "metrics directly calculated in Euclidean space" refers to variance and our newly added metrics (MeanAD, MedianAD, and IQR) mentioned in our previous response.

---

> > > ### Author Response · Authors · 2024-11-28
> > >
> > > **Q7**: Explanation of the value of the paper, including but not limited to its applications and subsequent research.
> > >
> > > **A7**: Thanks for your question! We summarize the value of our paper below:
> > >
> > > 1. **Our proposed metric could promote subsequent research of error slice discovery by providing convenient evaluation and comparisons for error slice discovery algorithms without relying on predefined metadata or human involvement**: Previous evaluation of error slice discovery algorithms typically rely on: metrics that require access to metadata, which is unavailable or scarce in many scenarios; Or investigation of concrete examples sampled from error slices, requiring human involvement. With our proposed coherence metric, we can directly evaluate algorithms of error slice discovery without predefined and labeled metadata or human involvement. This is similar to FID (Fréchet inception distance) serving as a metric of evaluating fidelity of image generation methods without requiring user studies. Thus our metric could promote subsequent research on error slice discovery, which is a relatively new and developing field.
> > > 2. **Our algorithm has the potential to be extended to various tasks or data types**: Previous algorithms require prediction probabilities as input, thus limited in classification tasks. In contrast, our algorithm requires prediction loss instead of prediction probabilities and is naturally flexible to be applied to various tasks, which we demonstrate through case studies of various tasks in our paper. In future work, we could explore our algorithm's potential to be extended to more types of tasks, e.g. more specific tasks of images or text like object tracking, word segmentation, etc, or tasks of other data types like audio, graph, etc.
> > > 3. **The area of error slice discovery is of potential paramount importance but less investigated, which could deserve more attention and promotion**: There have been numerous powerful models with tremendous performance across a large number of tasks. However, they could still fail on certain samples or subpopulations, which greatly prevents their application in risk-sensitive areas like medical images and autonomous driving, where mistakes of models could lead to catastrophic consequences. Therefore, it is beneficial to understand failures of models so that we could remedy for them. Compared with investigating isolated failure cases, we pay more attention to identifying coherent and interpretable failure patterns shared by many samples or a subpopulation, which is generally more useful. On one hand, with an interpretable failure pattern identified, we could directly collect data based on the pattern to improve model performance in a data-centric way. On the other hand, we could even simply avoid using the models in the identified risky scenarios to ensure their usefulness and reliability. Thus we believe this problem and direction deserves more attention and promotion.
> > >
> > >
> > >
> > > ### References
> > >
> > > [1] Tenenbaum, Joshua B., Vin de Silva, and John C. Langford. "A global geometric framework for nonlinear dimensionality reduction." *Science* 290.5500 (2000): 2319-2323.
> > >
> > > [2] Belkin, Mikhail, and Partha Niyogi. "Laplacian eigenmaps for dimensionality reduction and data representation." *Neural computation* 15.6 (2003): 1373-1396.
> > >
> > > [3] Ma, Yunqian, and Yun Fu. "Manifold learning theory and applications." Vol. 434. Boca Raton: CRC press, 2012.
> > >
> > > [4] Chen, Dong, and Hans-Georg Müller. "Nonlinear manifold representations for functional data." *The Annals of Statistics* (2012): 1-29.
> > >
> > > [5] Melas-Kyriazi, Luke. "The mathematical foundations of manifold learning." *arXiv preprint arXiv:2011.01307* (2020).

---

> > > > ### Comment · Reviewer_q2Dt · 2024-11-29
> > > >
> > > > Thanks for the positive and valuable response, I have increased my rating.

---

> > > > > ### Author Response · Authors · 2024-11-29
> > > > >
> > > > > Thanks for your recognition of our work!

---

### Official Review · Reviewer_GUGx · 2024-11-02

**Soundness:** 3
**Presentation:** 2
**Contribution:** 3
**Rating:** 6
**Confidence:** 3

**Summary:**

This paper studies ERROR SLICE DISCOVERY issue. In the paper, the authors mainly proposed the concept of manifold compactness and a coherence metric. The proposed optimization objective can be applied to models of various tasks. The authors also conduct many experiments to demonstrate the effectiveness of the method.

**Strengths:**

The authors mainly proposed the concept of manifold compactness and a coherence metric.

**Weaknesses:**

The paper is difficult to understand and follow as the source code is not provided.

**Questions:**

1. The dimensions of all variables are not provided clearly, making the paper difficult to understand.
2. What’s the physical meaning of the second term in (2)? What’s the optimization solution for (2)?
3. There are not enough comparison methods.
4. What are the main advantages of the proposed method compared with the existing works?

---

> ### Author Response · Authors · 2024-11-22
>
> Thanks for your valuable feedback! We try to address your concerns below.
>
> **Q1-1**: The paper is difficult to understand and follow as the source code is not provided.
>
> **A1-1**: Thanks for your reminder. Now we add the **anonymous link to our code** in our **second common response**.
>
> **Q1-2**: The dimensions of all variables are not provided clearly, making the paper difficult to understand.
>
> **A1-2**: Thanks for your feedback. We would like to clarify that the dimensions of the main variables have been provided in Appendix A.1 in both versions of our paper, e.g. the feature dimension in our main experiments is 512, corresponding to $z_i$ in our main paper. For other variables without mentioning the dimensions, e.g. $X\in\mathcal{X}$ and $Y\in\mathcal{Y}$, we believe that they do not affect the understanding of our paper. We are willing to provide further explanations if there are remaining concerns on this.
>
> **Q2-1**: What’s the physical meaning of the second term in (2)?
>
> **A2-1**: Thanks for your question. The second term is our proposed manifold compactness (Definition 1) on a set of weighted samples. As stated in Line 255-257, for the convenience of optimization, we assign a sample weight $w_i$ for each data point and treat the sample weights as the variable to be optimized. Thus the first term in (2) $\sum_{i=1}^n w_il_i$ becomes the weighted loss, and the second term $\sum_{i=1}^n\sum_{j=1}^n w_iw_jq_{ij}$ in (2) becomes the manifold compactness on a set of weighted samples.
>
> **Q2-2**: What’s the optimization solution for (2)?
>
> **A2-2**: Thanks for your question. To optimize it, since it formulates a non-convex quadratic programming problem, as stated in Line 270-271, we employ the mathematical optimization solver Gurobi, which is one of the most popular and widely used solvers for standard mathematical programming problems like (2). Meanwhile, in the revised version of our paper, we add some theoretical analyses of its solution, which could be referred to in our **first common response**.
>
> **Q3**: There are not enough comparison methods.
>
> **A3**: Thanks for your feedback. We would like to remind that the problem of error slice discovery is relatively new and less investigated, which does not have many comparable methods. As stated in the section of related work (Appendix B), there are now two paradigms of error slice discovery methods, and we compare with methods of the first paradigm, which separates error slice discovery and the later interpretation. Methods of the second paradigm incorporate the discovery and interpretation of error slices together and their effectiveness heavily relies on the usage of large multi-modal models and text-to-image models, which may not be realiable sometimes. Thus they are inappropriate to be compared with methods of the first paradigm, which our method belongs to.
>
> **Q4**: What are the main advantages of the proposed method compared with the existing works?
>
> **A4**: Thanks for your question. First, the effectiveness of our method is directly associated with the validity of our proposed metric, which is inspired by the data geometry property and whose validity has been demonstrated in our preliminary studies in Section 3. Second, unlike previous methods that indirectly incorporate the constraint of coherence into their design, our method directly treats the constraint of coherence as a part of the optimization objective. Besides, most previous methods require prediction probability as input while our method only requires prediction loss, so our method is naturally flexible to be applied to different tasks while most previous methods can be only applied to classification. The superiority and flexibility of our method are empirically shown in our experiments on the benchmark and the case studies of various tasks.

---

> > ### Comment · Reviewer_GUGx · 2024-11-26
> >
> > Thanks for the responses. The responses address most of my concerns.

---

> > > ### Author Response · Authors · 2024-11-26
> > >
> > > Thanks for your response and efforts! We would appreciate it if you could provide details of the unresolved concerns, and we are very willing to engage in further discussions. We also sincerely hope that you might take the revisions mentioned in our common response to all reviewers into account when evaluating our contributions, including the theoretical analyses of our optimization objective and the newly added comparisons with metrics directly calculated in Euclidean space.

---

### Author Response · Authors · 2024-11-22
**Common Response**

We thank all reviewers for their valuable comments. In this common response, we will list all the changes we have made in the revised version of paper according to the reviews. The changes are also marked in blue on page 1, 5-6, 16, and 34-36.

1. On page 1, we modify Section 1 to better illustrate the motivation of error slice discovery in the first paragraph.
2. On page 5-6, we remove the training of the slicing function from the algorithm procedure and put it into the section of experiments as part of the experimental details, since we designed this component mainly for fair comparisons with previous methods that output a slicing function and it is not a necessary component of our algorithm. We also add a new subsection of Appendix A.11 on page 36 to illustrate that our method is still effective without the additionally trained slicing function.
3. On page 16, in Appendix A.1.2, we add comparisons with three other metrics that are directly calculated in the Euclidean space: Mean Absolute Deviation, Median Absolute Deviation, and Interquartile Range, so that the superiority of our proposed metric can be better demonstrated.
4. On page 34-35, we add a new subsection of Appendix A.10, where we conduct some theoretical analyses on Equation 2, the optimization objective of our method. We theoretically prove that our optimization objective not only explicitly considers the manifold compactness inside the identified slice, but also implicitly considers the separability between samples in and out of the identified slice. Meanwhile, we theoretically prove that our optimization objective is equivalent to the discrete version of sample selection with proper assumptions, which confirms the validity of transforming the sample selection problem into the continuous version for the convenience of optimization.

We sincerely hope that these modifications could help address reviewers' concerns, and that reviewers might kindly take these improvements into account when evaluating our contributions after rebuttal. Thanks again for your time and efforts in reviewing our work!

---

> ### Author Response · Authors · 2024-11-28
> **Minor modifications**
>
> We thank all reviewers again for their valuable feedback and responses!
>
> As suggested by Reviewer q2Dt, to be rigorous, we make minor modifications in terms of expressions related to Euclidean distance/space in our second revision of paper, which we mark in orange on page 2-4 and 15-16.

---

### Meta-Review · Area_Chair_bFSN · 2024-12-13

**Metareview:**

I have read all the materials of this paper including the manuscript, appendix, comments, and response. Based on collected information from all reviewers and my personal judgment, I can make the recommendation on this paper, reject. No objection from reviewers who participated in the internal discussion was raised against the reject recommendation.

**Research Questions**

This paper considers error slice discovery problem that identifies several semantically coherent and incorrectly predicted subsets.

**Challenge Analysis**

The authors argue that the existing methods employ the predefined metadata and do not consider the coherence.

**Philosophy**

The authors aim to solve the research problem purely from the data and take the coherence into consideration.

**Technique**

Based on the above philosophy, the authors formulate an optimization. The technique is straightforward but suffers from high computational cost. Moreover, the authors do not consider the case with multiple error slice.

**Experiment**

The experiments are not solid, which do not well address the research problem. (1) The authors fail to demonstrate the identified subset is semantically coherent. The trivial solution that returns all the wrongly prediction samples as a set can achieve high performance in terms of the metrics used in this paper. (2) Parameter analysis and corresponding experiment setup is missing.

**Additional Comments On Reviewer Discussion:**

No objection from reviewers who participated in the internal discussion was raised against the reject recommendation.

---

### Decision · Program_Chairs · 2025-01-22

Reject